# QUADRATIC DIRECT FORECAST FOR TRAINING MULTI-STEP TIME-SERIES FORECAST MODELS

**Hao Wang**[1,2]  **Licheng Pan**[1]  **Yuan Lu**[1]  **Zhichao Chen**[3]  **Tianqiao Liu**[4]  **Shuting He**[5]
**Zhixuan Chu**[6]  **Qingsong Wen**[7]  **Haoxuan Li**[8†]  **Zhouchen Lin**[3,9†]

[1]Xiaohongshu Inc.
[2]College of Control Science and Technology, Zhejiang University
[3]State Key Lab of General AI, School of Intelligence Science and Technology, Peking University
[4]College of Engineering, Purdue University
[5]School of Computing and Artificial Intelligence, Shanghai University of Finance and Economics
[6]College of Computer Science and Technology, Zhejiang University
[7]Squirrel AI    [8]Center for Data Science, Peking University
[9]Institute for Artificial Intelligence, Peking University

## ABSTRACT

The design of learning objectives is central to training time-series forecasting models. Existing learning objectives such as mean squared error mostly treat each future step as an independent, equally weighted task, which leads to the following two challenges: (1) they overlook the *label autocorrelation effect* among future steps, leading to biased learning objectives; (2) they fail to set *heterogeneous task weights* for different forecasting tasks corresponding to varying future steps, limiting the forecasting performance. To fill this gap, we propose a novel quadratic-form weighted learning objective, addressing both issues simultaneously. Specifically, the off-diagonal elements of the weighting matrix account for the label autocorrelation effect, whereas the non-uniform diagonals are expected to match the preferred weights of the forecasting tasks with varying future steps. On this basis, we propose a Meta-learning Direct Forecast (MetaDF) learning algorithm, which trains the forecast model using the adaptively updated quadratic-form weighting matrix. Experiments show that our MetaDF effectively improves the performance of various forecast models, achieving state-of-the-art results. Code is available at https://github.com/Master-PLC/QDF.

## 1 INTRODUCTION

Time-series forecasting, which involves predicting future values from past observations, is foundational to a wide range of applications, including meteorological prediction (Bi et al., 2023), financial stock forecasting (Li et al., 2025a), and robotic trajectory forecasting (Fan et al., 2023). In the context of deep learning, the development of robust forecasting models relies on two crucial components (Wang et al., 2025d): *(1) the design of neural architectures for forecasting and (2) the formulation of suitable learning objectives for model training.* Both present distinct challenges.

Recent research has focused intensively on the first aspect, namely, neural architecture design. The principal challenge lies in efficiently capturing the autocorrelation structures in the history sequence. A variety of architectures have been proposed (Wu et al., 2023; Luo and Wang, 2024; Gu et al., 2021). One exemplar would be Transformer models that employ self-attention to model autocorrelation and scale effectively (Liu et al., 2024; Nie et al., 2023; Piao et al., 2024). Another rapidly developing direction would be linear models, which use linear projections to model autocorrelation and demonstrate competitive performance (Lin et al., 2024; Zeng et al., 2023; Yue et al., 2025). These advances showcase the fast-paced evolution of model architectures for time-series forecasting.

In contrast, the formulation of learning objectives remains relatively underexplored (Li et al., 2025b; Qiu et al., 2025; Kudrat et al., 2025). Most recent studies resort to mean squared error (MSE)

---

[†]Corresponding authors.

as the learning objective (Lin et al., 2025; 2024; Liu et al., 2024). However, MSE overlooks the autocorrelation effect present in label sequences, which renders it a biased objective (Wang et al., 2025e;d). Additionally, it assigns equal weights to all forecasting tasks with varying future steps, ignoring the potential of a heterogeneous weighting scheme. As a result, the learning objective design of forecast models is challenged by the label autocorrelation effect and heterogeneous task weights, which are not fully addressed by existing methods.

In this work, we first propose a novel quadratic-form weighted learning objective that simultaneously tackles both issues. Specifically, the off-diagonal elements of the weighting matrix model the label autocorrelation effect, while the non-uniform diagonal elements enable the assignment of heterogeneous task weights to different future steps. Building on this, we introduce the Quadratic Direct Forecast (QDF) learning algorithm, which trains the forecasting model using an adaptively updated quadratic-form weighting matrix. Our main contributions are summarized as follows:

- We identify two fundamental challenges in designing learning objectives for time-series forecast models: the label autocorrelation effect and the heterogeneous task weights.

- We propose a quadratic-form weighted learning objective that tackles both challenges. The QDF learning algorithm is proposed to apply the objective for training time-series forecast models.

- We perform comprehensive empirical evaluations to demonstrate the effectiveness of QDF, which enhances the performance of state-of-the-art forecast models across diverse datasets.

## 2 PRELIMINARIES

### 2.1 PROBLEM DEFINITION

This work investigates the multi-step time-series forecasting task. Formally, given a time-series dataset $\boldsymbol{S}$ with D covariates, the history sequence at time step $n$ is denoted by $\boldsymbol{X} = [\boldsymbol{S}_{n-H+1}, \ldots, \boldsymbol{S}_n] \in \mathbb{R}^{H \times D}$, while the label sequence is $\boldsymbol{Y} = [\boldsymbol{S}_{n+1}, \ldots, \boldsymbol{S}_{n+T}] \in \mathbb{R}^{T \times D}$, where H and T denote the history and forecast horizons, respectively. Recent approaches predominantly adopt a direct forecasting (DF) paradigm, predicting all T future steps simultaneously (Liu et al., 2024; Piao et al., 2024). Therefore, the goal is to learn a parameterized model $g_\theta : \mathbb{R}^{H \times D} \to \mathbb{R}^{T \times D}$ that generates forecast sequence $\hat{\boldsymbol{Y}}$ approximating $\boldsymbol{Y}$, where $\theta$ represents the learnable parameters[1].

Advances in forecasting models typically revolve around two axes: (1) the design of neural architectures for encoding historical inputs (Liu et al., 2024; Zeng et al., 2023); and (2) the design of learning objectives for effective training (Wang et al., 2025d;e; Cuturi and Blondel, 2017; Sakoe and Chiba, 2003). This study is primarily concerned with the latter—specifically, the improved formulation of learning objectives. Nonetheless, we briefly introduce both aspects as follows for completeness.

### 2.2 NEURAL NETWORK ARCHITECTURES IN TIME-SERIES FORECASTING

The principal goal of architecture development in time-series forecasting is to learn informative representations of history sequence. The key challenge is to accommodate the autocorrelation effect present in the history sequence. Traditional approaches include recurrent neural networks (Gu et al., 2021; Chen et al., 2023), convolutional neural networks (Wu et al., 2023; Luo and Wang, 2024), and graph neural networks (Cao et al., 2020; Mateos et al., 2019). In the recent literature, one predominant series are Transformer models (e.g., TQNet (Lin et al., 2025), PatchTST (Nie et al., 2023), iTransformer (Liu et al., 2024)), which show strong scalability on large datasets but at a higher computational cost. Another predominant series are linear models (e.g., TimeMixer (Wang et al., 2024), DLinear (Zeng et al., 2023)), which are efficient but may struggle to scale and cope with varying history sequence length. There are also hybrid architectures that fuse Transformer and linear modules to combine their respective advantages (Lin et al., 2024; Wu et al., 2025).

---

[1]Hereafter, we consider the univariate case (D = 1) for clarity. In the multivariate case, each variable can be treated as a separate univariate case when computing the learning objectives.

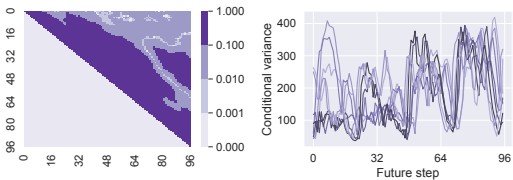 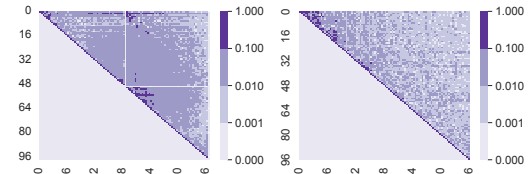

(a) Partial correlation and significance of labels.   (b) Partial correlation of extracted label components.

Figure 1: Statistics of label components conditioned on $\boldsymbol{X}$, with a forecast horizon of T = 96. (a) Partial correlation and conditional variance estimated from the raw label sequence $Y$, with colors indicating different $\boldsymbol{X}$. (b) Partial correlation matrices of label components extracted by FreDF and Time-o1 (Wang et al., 2025e;d). Calculation details are provided in Appendix A.

### 2.3 LEARNING OBJECTIVES IN TIME-SERIES FORECASTING

The primary challenge driving the development of learning objectives in time-series forecasting is to accommodate the autocorrelation effect present in the label sequence. The standard mean squared error (MSE) is widely used to train forecast models (Lin et al., 2025; 2024; Liu et al., 2024), which measures the point-wise difference between the forecast and label sequences:

$$\mathcal{L}_{\mathrm{mse}} = \|\boldsymbol{Y} - g_\theta(\boldsymbol{X})\|^2, \tag{1}$$

However, $\mathcal{L}_{\mathrm{mse}}$ is known to be biased, as it neglects the presence of autocorrelation in the label sequence (Wang et al., 2025e). To mitigate this issue, alternative objectives have been explored. One line of work promotes shape alignment between the forecast and label sequences (Le Guen and Thome, 2019; Cuturi and Blondel, 2017), emphasizing the autocorrelation structure, though these approaches generally lack theoretical guarantees for bias elimination. Another line of works transforms the labels into decorrelated components before alignment, thereby mitigating bias and improving forecast performance (Wang et al., 2025e;d). These empirical advancements underscore the critical role of objective function design in advancing time-series forecasting.

## 3 METHODOLOGY

### 3.1 MOTIVATION

The design of learning objectives is central to training time-series forecasting models. Likelihood maximization provides a principled approach, minimizing the negative log-likelihood (NLL) of label sequence. By Theorem 3.1, the NLL is a quadratic form weighted by the inverse of the conditional covariance matrix $\bar{\boldsymbol{\Sigma}}$. This formulation reveals two key challenges in designing learning objectives.

- **Autocorrelation effect.** Time-series data exhibit strong autocorrelation, where observations are highly correlated with their past values. This implies that future steps within the label sequence are correlated even when conditioned on the history $\boldsymbol{X}$ (Wang et al., 2025e). This property necessitates modeling the off-diagonal elements of $\bar{\boldsymbol{\Sigma}}$, which are not necessarily zeros.

- **Heterogeneous weights.** The training of forecast models is a typical multitask learning problem, where predicting each future step is a distinct task. These tasks often exhibit varying levels of difficulty and uncertainty, suggesting they require different weights during optimization. This property necessitates modeling the diagonal elements of $\bar{\boldsymbol{\Sigma}}$, which are not necessarily uniform.

**Theorem 3.1** (Likelihood formulation). *Given history sequence $\boldsymbol{X}$, let $\boldsymbol{Y} \in \mathbb{R}^{\mathrm{T}}$ be the associated label sequence and $g_\theta(\boldsymbol{X}) \in \mathbb{R}^{\mathrm{T}}$ be the forecast sequence. Assuming the forecast errors follow a multivariate Gaussian distribution, the NLL of the label sequence, omitting constant terms, is:*

$$\mathcal{L}_{\boldsymbol{\Sigma}}(\boldsymbol{X}, \boldsymbol{Y}; g_\theta) = \|\boldsymbol{Y} - g_\theta(\boldsymbol{X})\|_{\bar{\boldsymbol{\Sigma}}}^2 = (\boldsymbol{Y} - g_\theta(\boldsymbol{X}))^\top \bar{\boldsymbol{\Sigma}}(\boldsymbol{Y} - g_\theta(\boldsymbol{X})), \tag{2}$$

*where $\boldsymbol{\Sigma} \in \mathbb{R}^{\mathrm{T} \times \mathrm{T}}$ is the conditional covariance of the label sequence given $\boldsymbol{X}$, $\bar{\boldsymbol{\Sigma}}$ is the inverse of $\boldsymbol{\Sigma}$.*

However, it is infeasible to directly minimize $\mathcal{L}_{\bar{\boldsymbol{\Sigma}}}$ for model training. The conditional covariance $\boldsymbol{\Sigma}$ is unknown and intractable to estimate from the single label sequence typically available per $\boldsymbol{X}$.

This difficulty leads to the widespread adoption of the mean squared error (MSE) objective, which in essence assumes $\bar{\Sigma}$ is an identity matrix (Lin et al., 2025) and therefore fails to model either autocorrelation or heterogeneous uncertainty. Subsequent works advocate transforming the labels into latent components for alignment, exemplified by **FreDF** (Wang et al., 2025e) and **Time-o1** (Wang et al., 2025d). However, the transformations they employ guarantee only *marginal decorrelation* of the obtained components, not the required *conditional decorrelation* (i.e., diagonal $\bar{\Sigma})^2$, thereby failing to accommodate the autocorrelation effect. Moreover, they assign equal weight to optimize each component, thereby failing to accommodate heterogeneous weights. *Hence, existing methods fail to address the two challenges in designing learning objectives for time-series forecast models.*

**Case study.** We conducted a case study on the ECL dataset to substantiate our claims (Fig. 1). The primary observations are summarized as follows:

- **The identified challenges are prominent.** As shown in Fig. 1(a), the partial correlation matrix exhibits significant off-diagonal values (with over 61.4% exceeding 0.1), confirming the presence of autocorrelation effect. Additionally, the conditional variances differ considerably across future steps, highlighting the importance of using heterogeneous error weights.
- **Existing methods fail to fully address them.** The partial correlation coefficients of the latent components extracted by FreDF and Time-o1 (Wang et al., 2025e;d) are presented in Fig. 1(b). Although the non-diagonal elements are notably reduced, residual values remain, indicating that these methods do not completely eliminate autocorrelation in the transformed components.

Given the critical role of the weighting matrix in elucidating the two challenges and the limitation of existing methods, it is essential to investigate strategies for incorporating the weighting matrix into the design of learning objectives for training forecast models. Specifically, three key questions arise: *(1) How can the weighting matrix be estimated from data? (2) How to define a learning objective for model training with it? (3) Does it improve forecasting performance?*

## 3.2 LEARNING WEIGHTING MATRIX TARGETING GENERALIZATION

A direct approach to incorporating the weighting matrix $\bar{\Sigma}$ is to use the NLL from (2). However, as previously established, it is impractical for training because the true conditional covariance $\Sigma$ is unknown and intractable to estimate accurately from data. To overcome this challenge, we advocate to learn *proxy* $\Sigma$ targeting model generalization. To this end, we treat $\Sigma$ as learnable parameters and the associated optimization problem is formulated in Definition 3.2.

**Definition 3.2.** Let $\mathcal{D}_{\text{in}} = (\boldsymbol{X}_{\text{in}}, \boldsymbol{Y}_{\text{in}})$ and $\mathcal{D}_{\text{out}} = (\boldsymbol{X}_{\text{out}}, \boldsymbol{Y}_{\text{out}})$ be non-overlapping splits of the training data, each consisting of historical and label sequences. The bilevel optimization problem is

$$\min_{\boldsymbol{\Sigma} \succeq 0} \mathcal{L}_{\boldsymbol{\Sigma}} \left( \boldsymbol{X}_{\text{out}}, \boldsymbol{Y}_{\text{out}}; g_{\theta^*} \right) \quad \text{where} \quad \theta^\star = \arg \min_\theta \mathcal{L}_{\boldsymbol{\Sigma}}(\boldsymbol{X}_{\text{in}}, \boldsymbol{Y}_{\text{in}}; g_\theta). \tag{3}$$

where $\boldsymbol{\Sigma} \succeq 0$ means $\boldsymbol{\Sigma}$ is semi-definite positive, a fundamental property of covariance matrix.

There are two loops in the optimization problem (3). The inner problem trains the forecast model $g_\theta$ on a data split $\mathcal{D}_{\text{in}}$ using a fixed $\bar{\Sigma}$; the outer problem then updates $\bar{\Sigma}$ to improve the generalization performance of the trained model on a disjoint holdout set $\mathcal{D}_{\text{out}}$. This process ensures the learned $\bar{\Sigma}$ produces a learning objective that drives the forecast model to generalize well.

**Re-parameterization.** To solve the problem (3), it is crucial to enforce $\boldsymbol{\Sigma} \succeq 0$. We address this by reparameterizing $\boldsymbol{\Sigma}$ via its Cholesky factorization, $\boldsymbol{\Sigma} = \boldsymbol{L}\boldsymbol{L}^\top$, where $\boldsymbol{L}$ is a lower-triangular matrix with positive diagonals (which can be ensured with a softplus activation). This reparameterization converts the constrained optimization over $\boldsymbol{\Sigma}$ into an unconstrained optimization over $\boldsymbol{L}$, thus enabling the use of standard gradient-based optimization methods. For clarity, in the following derivations, we continue to use $\boldsymbol{\Sigma}$ and omit the notational complexity introduced by this reparameterization.

Algorithm 1 details the procedure for solving (3) via gradient descent. The algorithm begins by partitioning the dataset $\mathcal{D}$ into two disjoint subsets, $\mathcal{D}_{\text{in}}$ and $\mathcal{D}_{\text{out}}$ (step 1). Within the inner loop, the objective $\mathcal{L}_{\boldsymbol{\Sigma}}$ is evaluated over $\mathcal{D}_{\text{in}}$, and its gradient with respect to the model parameters $\theta$ drives the update of $\theta$ (steps 2-5). Subsequently, in the outer loop, the objective is evaluated on $\mathcal{D}_{\text{out}}$, and

---

[2]This property is demonstrated in Theorem 3.3 (Wang et al., 2025e) and Lemma 3.2 (Wang et al., 2025d).

| **Algorithm 1** Atomic update procedure of QDF. | **Algorithm 2** The overall workflow of QDF. |
|---|---|
| **Input**: $g_\theta$: forecast model, $\Sigma$: weighting matrix, $\mathcal{D}$: dataset used to learn $\Sigma$.
**Parameter**: N: number of updates, $\eta$: update rate.
**Output**: $\Sigma$: obtained weighting matrix. | **Input**: $g_\theta$: forecast model, $\mathcal{D}_{\text{train}}$: training set.
**Parameter**: $N_{\text{in}}$: round of inner update, $N_{\text{out}}$: round of outer update, $\eta$: update rate, K: number of splits.
**Output**: $\mathcal{L}$: obtained learning objective. |
| 1: $\mathcal{D}_{\text{in}}, \mathcal{D}_{\text{out}} \leftarrow \text{split}(\mathcal{D})$
2: **for** $n = 1, 2, ..., N$ **do**
3: $\quad X_{\text{in}}, Y_{\text{in}} \leftarrow \mathcal{D}_{\text{in}}$
4: $\quad \theta \leftarrow \theta - \nabla_\theta \mathcal{L}_\Sigma(X_{\text{in}}, Y_{\text{in}}; g_\theta)$
5: **end for**
6: $X_{\text{out}}, Y_{\text{out}} \leftarrow \mathcal{D}_{\text{out}}$
7: $\Sigma \leftarrow \Sigma - \nabla_\Sigma \mathcal{L}_\Sigma(X_{\text{out}}, Y_{\text{out}}; g_\theta)$ | 1: $\Sigma \leftarrow I_{\text{T}}, \quad \mathcal{D}_1, \mathcal{D}_2, ..., \mathcal{D}_{\text{K}} \leftarrow \text{split}(\mathcal{D}_{\text{train}})$
2: **while** $n = 1, 2, ..., N_{\text{out}}$ **do**
3: $\quad \Sigma_{n+1} \leftarrow \text{Algorithm1}(\Sigma_n, \mathcal{D}_k, g_\theta), k = 1, ..., \text{K}$
4: $\quad$ **if** $\|\Sigma_{n+1} - \Sigma_n\|_{\text{F}} < 1e^{-4}$: **break**.
5: **end while**
6: $X_{\text{train}}, Y_{\text{train}} \leftarrow \mathcal{D}_{\text{train}}$
7: $\mathcal{L} \leftarrow \mathcal{L}_{\Sigma_{n+1}}(X_{\text{train}}, Y_{\text{train}}; g_\theta)$ |

the gradient with respect to $\Sigma$ is used to update the weighting matrix (steps 6-7). It is important to emphasize that the gradient in the outer loop is backpropagated through the updated model parameters $\theta$ to $\Sigma$, rather than being taken directly with respect to $\Sigma$. This approach ensures that the impact of $\Sigma$ on the learned $\theta$—and thereby on the generalization performance—is fully captured. Collectively, the procedure above performs a single update of $\Sigma$ toward the optimum of (3), and can be repeatedly applied to iteratively refine the estimate of $\Sigma$.

### 3.3 THE WORKFLOW OF QDF FOR TRAINING TIME-SERIES FORECAST MODELS

While we have established a method to learn an instrumental weighting matrix $\Sigma$, it is not clear how to use the obtained $\Sigma$ for training forecast models. To fill this gap, we detail the workflow of QDF, which first learns $\Sigma$ and then applies it to train forecast models. The principal steps are encapsulated in Algorithm 2, which consists of three primary phases as follows.

- **Initialization.** The process begins by initializing $\Sigma$ as an identity matrix. The training set $\mathcal{D}_{\text{train}}$ is split chronologically into K non-overlapping subsets (step 1). This partitioning is crucial for robustness: by updating $\Sigma$ across different data distributions (subsets), we seek for an estimation of $\Sigma$ that is less likely to overfit to any single part of the training data (Nichol and Schulman, 2018).

- **Weighting matrix learning.** With the data prepared, we iteratively refine $\Sigma$ by applying Algorithm 1 sequentially across the K subsets. The iteration stops when $\Sigma$ converges (i.e., the change between iterations is negligible) or a predefined number of rounds is completed (steps 2-5).

- **Model training.** With the learned weighting matrix $\Sigma$ in hand, the final phase is to train the forecast model $g_\theta$. This is achieved by minimizing the corresponding NLL objective ($\mathcal{L}_\Sigma$) over the training set (steps 6-7). In practice, this minimization is performed using standard gradient descent, and the NLL objective can be estimated on mini-batches for computational efficiency.

By employing $\mathcal{L}_\Sigma$ for model training, QDF effectively leverages the weighting matrix $\Sigma$, thereby addressing the two established challenges. Specifically, the off-diagonal elements of $\Sigma$ enable the model to account for the autocorrelation effect, and non-uniform diagonals enable heterogeneous weights for each error term. There is no risk of data leakage, as the full training procedure (Algorithm 2) exclusively utilizes the training set, without using the validation or test sets. Notably, QDF is model-agnostic, making it a versatile tool for improving the training of various direct forecast models (Liu et al., 2024; Zeng et al., 2023; Piao et al., 2024).

The strategy of treating $\Sigma$ as learnable parameters is conceptually related to the principles of meta-learning (Nichol et al., 2018; Finn et al., 2017). However, our work diverges from meta-learning in both goal and implementation. (1) The goal of meta-learning is to enable rapid adaptation to new, dynamic tasks, whereas QDF is designed to construct a static objective for time-series forecasting—specifically accommodating autocorrelation and heterogeneous weights. (2) This difference in goals leads to different validation schemes. Meta-learning validates generalization on a set of new tasks, whereas QDF uses a holdout dataset drawn from the same forecasting task for validation. (3) In time-series analysis, some studies accommodate meta-learning for model selection (Talagala et al., 2023), ensembling (Montero-Manso et al., 2020), initialization (Oreshkin

Table 1: Long-term forecasting performance.

| Models | QDF (Ours) | | TQNet (2025) | | PDF (2024) | | Fredformer (2024) | | iTransformer (2024) | | FreTS (2023) | | TimesNet (2023) | | MICN (2023) | | TiDE (2023) | | PatchTST (2023) | | DLinear (2023) | |
|---|---|---|---|---|---|---|---|---|---|---|---|---|---|---|---|---|---|---|---|---|---|---|
| Metrics | MSE | MAE | MSE | MAE | MSE | MAE | MSE | MAE | MSE | MAE | MSE | MAE | MSE | MAE | MSE | MAE | MSE | MAE | MSE | MAE | MSE | MAE |
| ETTm1 | **0.371** | **0.389** | _0.376_ | _0.391_ | 0.387 | 0.396 | 0.387 | 0.398 | 0.411 | 0.414 | 0.414 | 0.421 | 0.438 | 0.430 | 0.396 | 0.421 | 0.413 | 0.407 | 0.389 | 0.400 | 0.403 | 0.407 |
| ETTm2 | **0.270** | **0.317** | _0.277_ | _0.321_ | 0.283 | 0.331 | 0.280 | 0.324 | 0.295 | 0.336 | 0.316 | 0.365 | 0.302 | 0.334 | 0.308 | 0.364 | 0.286 | 0.328 | 0.303 | 0.344 | 0.342 | 0.392 |
| ETTh1 | **0.431** | **0.431** | 0.449 | 0.439 | 0.452 | 0.440 | _0.447_ | _0.434_ | 0.452 | 0.448 | 0.489 | 0.474 | 0.472 | 0.463 | 0.533 | 0.519 | 0.448 | 0.435 | 0.459 | 0.451 | 0.456 | 0.453 |
| ETTh2 | **0.368** | **0.397** | _0.375_ | 0.400 | 0.375 | _0.399_ | 0.377 | 0.402 | 0.386 | 0.407 | 0.524 | 0.496 | 0.409 | 0.420 | 0.620 | 0.546 | 0.378 | 0.401 | 0.390 | 0.413 | 0.529 | 0.499 |
| ECL | **0.165** | **0.257** | _0.175_ | _0.265_ | 0.198 | 0.281 | 0.191 | 0.284 | 0.179 | 0.270 | 0.199 | 0.288 | 0.212 | 0.306 | 0.192 | 0.302 | 0.215 | 0.292 | 0.195 | 0.286 | 0.212 | 0.301 |
| Weather | **0.242** | **0.268** | _0.246_ | _0.270_ | 0.265 | 0.283 | 0.261 | 0.282 | 0.269 | 0.289 | 0.249 | 0.293 | 0.271 | 0.295 | 0.264 | 0.321 | 0.272 | 0.291 | 0.267 | 0.288 | 0.265 | 0.317 |
| PEMS03 | **0.089** | **0.197** | 0.119 | _0.217_ | 0.181 | 0.286 | 0.146 | 0.260 | 0.122 | 0.233 | 0.149 | 0.261 | 0.126 | 0.230 | _0.106_ | 0.223 | 0.316 | 0.370 | 0.170 | 0.282 | 0.216 | 0.322 |
| PEMS08 | **0.120** | **0.221** | _0.139_ | _0.240_ | 0.210 | 0.301 | 0.171 | 0.271 | 0.149 | 0.247 | 0.174 | 0.275 | 0.152 | 0.243 | 0.153 | 0.258 | 0.318 | 0.378 | 0.201 | 0.303 | 0.249 | 0.332 |

*Note*: We fix the input length as 96 following Liu et al. (2024). **Bold** and _underlined_ denote best and second-best results, respectively. The reported results are averaged over forecast horizons: T=96, 192, 336 and 720. QDF employs the top-performing TQNet as the forecast model.

et al., 2021) and domain adaptation (Narwariya et al., 2020), whereas QDF aims to obtain a versatile learning objective. To our knowledge, this is a technically innovative strategy.

## 4 EXPERIMENTS

To demonstrate the efficacy of QDF, there are six aspects that deserve empirical investigation:

1. **Performance:** *How does QDF perform?* We compare the forecast performance of QDF against state-of-the-art baselines (Section 4.2) and learning objectives (Section 4.3).

2. **Gains:** *What makes it effective?* We perform an ablation study (Section 4.4) to investigate the contribution of each technical element to its overall performance.

3. **Versatility:** *Does it benefit different forecast models?* We compare the performance of DF and QDF using different forecast models (Section 4.5), with further results provided in Appendix D.4.

4. **Flexibility:** *Does the weighting matrix accommodate meta-learning methods?* We attempt to learn the weighting matrix using established meta-learning methods (Section 4.5).

5. **Sensitivity:** *Is it sensitive to hyperparameters?* We conduct a sensitivity analysis (Section 4.7) to show that its effectiveness across a wide range of hyperparameter values.

6. **Complexity:** *Is it computational expensive?* We investigate the running time of QDF given different settings (Appendix D.7).

### 4.1 SETUP

**Datasets.** Our experiments are conducted on public datasets for time-series forecasting, consistent with prior works (Wu et al., 2023; Liu et al., 2024). The employed datasets include: ETT (consisting of ETTh1, ETTh2, ETTm1, and ETTm2), Electricity (ECL), Weather, and PEMS. For each dataset, we adopt a standard chronological split into training, validation, and testing partitions. Further details on dataset statistics are available in Appendix C.1.

**Baselines.** We compare QDF with 10 previous methods, which we categorize into two groups (Wang et al., 2025d): (1) Transformer-based models: PatchTST (Nie et al., 2023), iTransformer (Liu et al., 2024), Fredformer (Piao et al., 2024), PDF (Dai et al., 2024) and TQNet (Lin et al., 2025); (2) Non-transformer based models: DLinear (Zeng et al., 2023), TiDE (Das et al., 2023), MICN (Wang et al., 2023b), TimesNet (Wu et al., 2023) and FreTS (Yi et al., 2023).

**Implementation.** To ensure a fair evaluation, all baseline models are reproduced using the official codebases (Lin et al., 2025). We train all models with the Adam optimizer (Kingma and Ba, 2015) to minimize MSE on the training set. Notably, we disable the *drop-last* trick during both training and inference to prevent data leakage and ensure fair comparisons, as suggested by Qiu et al. (2024). More implementation details are available in Appendix C.

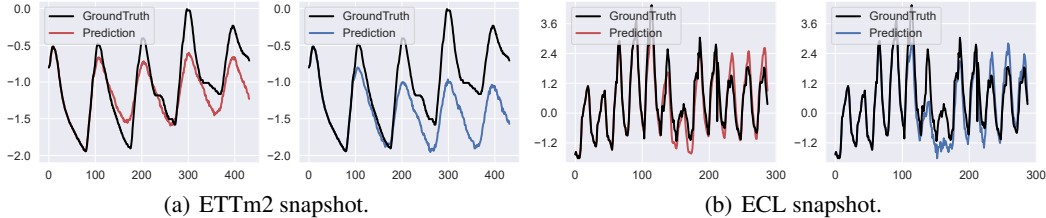

(a) ETTm2 snapshot.      (b) ECL snapshot.

Figure 2: The forecast sequence of DF (in blue) and QDF (in red), with historical length H = 96.

Table 2: Comparable results with other objectives for time-series forecast.

| Loss | | **QDF** | | Time-o1 | | FreDF | | Koopman | | Soft-DTW | | DF | |
|---|---|---|---|---|---|---|---|---|---|---|---|---|---|
| Metrics | | MSE | MAE | MSE | MAE | MSE | MAE | MSE | MAE | MSE | MAE | MSE | MAE |
| TQNet | ETTm1 | **0.371** | **0.389** | 0.372 | 0.390 | 0.375 | 0.390 | 0.595 | 0.499 | 0.387 | 0.394 | 0.376 | 0.391 |
| | ETTh1 | **0.431** | **0.431** | 0.437 | 0.432 | 0.432 | 0.432 | 0.451 | 0.442 | 0.453 | 0.438 | 0.449 | 0.439 |
| | ECL | **0.165** | **0.257** | 0.167 | 0.257 | 0.168 | 0.257 | 0.166 | 0.258 | 0.623 | 0.524 | 0.175 | 0.265 |
| | Weather | **0.242** | **0.268** | 0.245 | 0.269 | 0.244 | 0.268 | 0.282 | 0.306 | 0.255 | 0.276 | 0.246 | 0.270 |
| PDF | ETTm1 | **0.381** | **0.394** | 0.386 | 0.399 | 0.387 | 0.400 | 0.587 | 0.485 | 0.396 | 0.404 | 0.387 | 0.396 |
| | ETTh1 | **0.436** | **0.429** | 0.438 | 0.438 | 0.437 | 0.435 | 0.497 | 0.472 | 0.447 | 0.447 | 0.452 | 0.440 |
| | ECL | **0.194** | 0.277 | 0.195 | 0.276 | 0.194 | **0.274** | 0.196 | 0.281 | 0.695 | 0.548 | 0.198 | 0.281 |
| | Weather | **0.259** | **0.281** | 0.264 | 0.284 | 0.268 | 0.287 | 0.268 | 0.290 | 1.296 | 0.452 | 0.265 | 0.283 |

*Note*: **Bold** and underlined denote best and second-best results, respectively. The reported results are averaged over forecast horizons: T=96, 192, 336 and 720.

## 4.2 OVERALL PERFORMANCE

In this section, we compare the long-term forecasting results. As shown in Table 1, integrating QDF yields consistent improvements in forecast accuracy across all evaluated datasets. For instance, on the PEMS08 dataset, QDF achieves a notable reduction in both MSE and MAE by 0.019. We attribute the enhanced performance to QDF's adaptive weighting mechanism, which addresses two critical challenges in objective design: label autocorrelation effect and heterogeneous task weights.

**Examples.** A qualitative comparison between forecasts generated by DF versus QDF is presented in Fig. 2. The model trained with DF captures general patterns, but it often fails to model subtle dynamics. For example, on ETTm2, it struggles to follow a sustained upward trend, and on ECL, it misses a periodic peak around the 150th step. In contrast, QDF accurately captures these subtle patterns, which showcases its practical utility to improve real-world forecast performance.

## 4.3 LEARNING OBJECTIVE COMPARISON

In this section, we compare QDF against alternative learning objectives. Each objective is integrated into two forecast models: TQNet and PDF, using their official implementations. The results are summarized in Table 2. Overall, methods designed to correct for bias in likelihood estimation, namely FreDF and Time-o1, deliver consistent performance improvements. However, as we established in Section 3.1, these approaches cannot handle the two challenges and yield suboptimal performance. In contrast, QDF achieves the best performance, with its weighting matrix effectively tackling the two main challenges in objective design: the label autocorrelation effect and heterogeneous task weights.

## 4.4 ABLATION STUDIES

In this section, we examine the technical components within QDF that address the two key challenges of learning objective design and assess their individual contributions to forecast performance. The results are presented in Table 3, with key observations as follows:

- QDF† enhances DF by enabling heterogeneous task weights. Specifically, this variant follows the QDF procedure but sets the off-diagonal elements of the weighting matrix to zero while allowing

Table 3: Ablation study results.

| Model | Hetero. | Auto. | Data | T=96 MSE | T=96 MAE | T=192 MSE | T=192 MAE | T=336 MSE | T=336 MAE | T=720 MSE | T=720 MAE | Avg MSE | Avg MAE |
|---|---|---|---|---|---|---|---|---|---|---|---|---|---|
| DF | ✗ | ✗ | ETTm1 | 0.310 | 0.352 | 0.356 | 0.377 | 0.388 | 0.400 | 0.450 | 0.437 | 0.376 | 0.391 |
| | | | ETTh1 | 0.372 | 0.391 | 0.430 | 0.424 | 0.486 | 0.454 | 0.507 | 0.486 | 0.449 | 0.439 |
| | | | ECL | 0.143 | 0.237 | 0.161 | 0.252 | 0.178 | 0.270 | 0.218 | 0.303 | 0.175 | 0.265 |
| | | | Weather | 0.160 | 0.203 | 0.210 | 0.247 | 0.267 | 0.289 | 0.346 | 0.342 | 0.246 | 0.270 |
| QDF† | ✓ | ✗ | ETTm1 | 0.309 | 0.351 | 0.354 | 0.378 | 0.387 | 0.401 | 0.450 | 0.439 | 0.375 | 0.392 |
| | | | ETTh1 | 0.372 | 0.394 | 0.432 | 0.424 | _0.475_ | **0.445** | 0.494 | 0.481 | 0.443 | 0.436 |
| | | | ECL | _0.135_ | _0.230_ | 0.154 | 0.246 | _0.170_ | _0.263_ | 0.203 | 0.293 | _0.166_ | _0.258_ |
| | | | Weather | _0.159_ | _0.202_ | _0.208_ | _0.246_ | _0.265_ | _0.287_ | 0.344 | 0.341 | _0.244_ | _0.269_ |
| QDF‡ | ✗ | ✓ | ETTm1 | _0.308_ | _0.351_ | _0.353_ | _0.377_ | _0.385_ | _0.399_ | _0.443_ | _0.436_ | _0.372_ | _0.391_ |
| | | | ETTh1 | _0.369_ | _0.391_ | _0.430_ | _0.422_ | 0.477 | _0.447_ | _0.492_ | _0.475_ | _0.442_ | _0.434_ |
| | | | ECL | 0.136 | 0.230 | _0.153_ | _0.245_ | 0.171 | 0.264 | _0.203_ | _0.292_ | 0.166 | 0.258 |
| | | | Weather | 0.159 | 0.202 | 0.210 | 0.247 | 0.266 | 0.289 | _0.343_ | _0.340_ | 0.245 | 0.269 |
| QDF | ✓ | ✓ | ETTm1 | **0.307** | **0.349** | **0.352** | **0.376** | **0.383** | **0.398** | **0.441** | **0.434** | **0.371** | **0.389** |
| | | | ETTh1 | **0.365** | **0.389** | **0.427** | **0.421** | **0.466** | 0.449 | **0.466** | **0.467** | **0.431** | **0.431** |
| | | | ECL | **0.135** | **0.229** | **0.153** | **0.245** | **0.169** | **0.262** | **0.202** | **0.291** | **0.165** | **0.257** |
| | | | Weather | **0.158** | **0.201** | **0.207** | **0.245** | **0.263** | **0.286** | **0.342** | **0.339** | **0.242** | **0.268** |

*Note*: **Bold** and underlined denote best and second-best results, respectively. "Hetero." and "Auto." are abbreviations for heterogeneous task weight and label autocorrelation effect, respectively. It employs the top-performing TQNet as the forecast model. *Avg* indicates average results over forecast horizons: T=96, 192, 336 and 720.

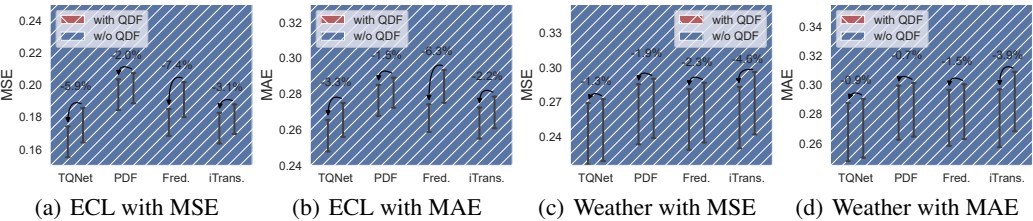

(a) ECL with MSE     (b) ECL with MAE     (c) Weather with MSE     (d) Weather with MAE

Figure 3: Improvement of QDF applied to different forecast models, shown with colored bars for means over forecast lengths (96, 192, 336, 720) and error bars for 50% confidence intervals.

the diagonal elements to be learned. It consistently outperforms DF, indicating that assigning heterogeneous weights to different forecast tasks can improve performance.

- QDF‡ improves DF by modeling label autocorrelation effects. Specifically, it fixes the diagonal elements of the weighting matrix to one, while learning the off-diagonal elements. It also surpasses DF, achieving the second-best results overall. This highlights the benefit of modeling autocorrelation effects in the learning objective for forecasting performance.

- QDF integrates both factors above and achieves the best performance, demonstrating the synergistic effect of addressing both heterogeneous task weights and label autocorrelation.

## 4.5 GENERALIZATION STUDIES

In this section, we explore the versatility of QDF as a model-agnostic enhancement. To this end, we integrate it into different forecast models: TQNet, PDF, FredFormer and iTransformer. The results in Fig. 3 show that QDF delivers consistent performance gains across all evaluated models. For example, on the ECL dataset, augmenting FredFormer and TQNet with QDF reduced their MSE by 7.4% and 5.9%, respectively. This consistent ability to elevate the performance of various models underscores QDF's versatility for improving time-series forecast performance.

## 4.6 FLEXIBILITY STUDIES

In this section, we explore the flexible implementation of QDF. Since the weighting matrix in QDF is treated as a set of learnable parameters, it is natural to investigate whether established meta-learning algorithms can be used to optimize it. To this end, we examine several representative meta-learning methods, including MAML (Finn et al., 2017), iMAML (Rajeswaran et al.,

Table 4: Comparison with meta-learning methods on ECL dataset.

| Method | T=96 | | T=192 | | T=336 | | T=720 | |
|---|---|---|---|---|---|---|---|---|
| | MSE | MAE | MSE | MAE | MSE | MAE | MSE | MAE |
| DF | 0.143 | 0.237 | 0.161 | 0.252 | 0.178 | 0.270 | 0.218 | 0.303 |
| iMAML | $0.135_{5.74\%\downarrow}$ | $0.230_{3.26\%\downarrow}$ | $\underline{0.154}_{4.31\%\downarrow}$ | $\underline{0.246}_{2.55\%\downarrow}$ | $0.170_{4.48\%\downarrow}$ | $0.263_{2.47\%\downarrow}$ | $0.205_{5.90\%\downarrow}$ | $0.293_{3.36\%\downarrow}$ |
| MAML | $0.136_{5.54\%\downarrow}$ | $0.230_{3.20\%\downarrow}$ | $0.154_{4.24\%\downarrow}$ | $0.246_{2.47\%\downarrow}$ | $0.170_{4.71\%\downarrow}$ | $0.263_{2.56\%\downarrow}$ | $0.205_{5.65\%\downarrow}$ | $0.293_{3.09\%\downarrow}$ |
| MAML++ | $\underline{0.135}_{5.76\%\downarrow}$ | $\underline{0.229}_{3.33\%\downarrow}$ | $0.154_{4.22\%\downarrow}$ | $0.246_{2.49\%\downarrow}$ | $\underline{0.170}_{4.72\%\downarrow}$ | $\underline{0.263}_{2.65\%\downarrow}$ | $\underline{0.204}_{6.41\%\downarrow}$ | $\underline{0.292}_{3.67\%\downarrow}$ |
| Reptile | $0.136_{5.06\%\downarrow}$ | $0.230_{2.90\%\downarrow}$ | $0.155_{3.73\%\downarrow}$ | $0.247_{2.14\%\downarrow}$ | $0.171_{3.91\%\downarrow}$ | $0.264_{2.07\%\downarrow}$ | $0.206_{5.36\%\downarrow}$ | $0.294_{2.96\%\downarrow}$ |
| QDF | $\mathbf{0.135}_{6.10\%\downarrow}$ | $\mathbf{0.229}_{3.63\%\downarrow}$ | $\mathbf{0.153}_{4.76\%\downarrow}$ | $\mathbf{0.245}_{2.82\%\downarrow}$ | $\mathbf{0.169}_{5.14\%\downarrow}$ | $\mathbf{0.262}_{2.71\%\downarrow}$ | $\mathbf{0.202}_{7.37\%\downarrow}$ | $\mathbf{0.290}_{4.09\%\downarrow}$ |

*Note*: **Bold** and underlined denote best and second-best results, respectively. The subscript denotes the relative error reduction compared with DF.

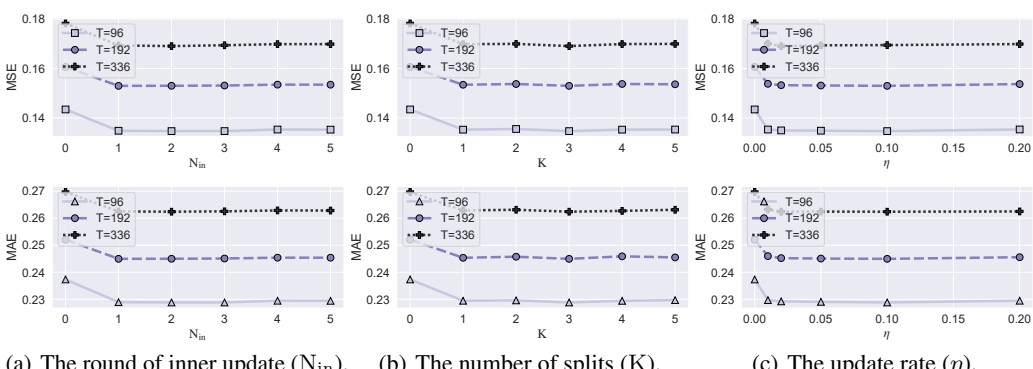

(a) The round of inner update ($N_{in}$).    (b) The number of splits (K).    (c) The update rate ($\eta$).

Figure 4: Impact of hyperparameters on the performance of QDF.

2019), MAML++(Antoniou et al., 2019), and Reptile(Nichol and Schulman, 2018). Overall, all these methods outperform the canonical DF approach that sets the weighting matrix as an identity matrix, thereby demonstrating the flexibility of QDF's implementation. However, these methods do not explicitly optimize the weighting matrix for out-of-sample generalization, which is a distinct advantage of our implementation that benefits forecast performance.

## 4.7 HYPERPARAMETER SENSITIVITY

In this section, we examine the impact of key hyperparameters on QDF's performance, with results shown in Fig. 4. The main observations are as follows:

- The coefficient $N_{in}$ determines the number of inner-loop updates in Algorithm 2. We observe that increasing $N_{in}$ from 0 to 1 significantly improves forecasting accuracy. Further increases bring marginal gains, suggesting that the forecast model's performance after one-step update already provides valuable signals to guide the weighting matrix update.

- The coefficient K determines the number of data splits in Algorithm 2. The best performance is achieved when K = 3, indicating that splitting the data enhances the generalization ability of the learned weighting matrix. Increasing it further leads to diminishing returns, as the sample size per split becomes too small to be informative given large values of K.

- The coefficient $\eta$ determines the update rate in Algorithm 2, where setting it to zero immediately reduces the method to the DF baseline. In general, using $\eta > 0$ to update the weighting matrix effectively improves performance, and the improvement is robust to a wide range of $\eta$ values.

## CONCLUSION

In this study, we identify two key challenges in designing learning objectives for forecast models: the label autocorrelation effect and heterogeneous task weights. We show that existing methods fail to address both challenges, resulting in suboptimal performance. To fill this gap, we introduce a novel quadratic-form weighted learning objective that simultaneously tackles these issues. To apply this

objective, we propose a QDF learning algorithm, which trains the forecast model using the quadratic objective with an adaptively updated weighting matrix. Experimental results demonstrate that QDF consistently enhances the performance of various forecasting models.

***Limitations & future works.*** A limitation of the current QDF is that it assumes a static weighting matrix $\bar{\Sigma}$ suffices to enhance the learning objective. While this assumption is well motivated to address the two identified challenges—label autocorrelation and heterogeneous task weighting—it constrains the expressiveness of the learned objective. A promising solution for future work would be to employ a hyper-network that dynamically generates input-dependent weighting matrix, thereby yielding a more adaptive and expressive formulation that could potentially lead to further performance improvements.

### ACKNOWLEDGMENTS

Z. Lin was supported by the NSF China (No. 62276004), the Beijing Natural Science Foundation (No. L257007), and the Beijing Major Science and Technology Project (No. Z251100008425006).

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

## A  ON THE LABEL AUTOCORRELATION ESTIMATION DETAILS

In this section, we introduce the procedure for estimating the label autocorrelation in Fig. 1. A primary challenge in this estimation is accounting for the confounding influence of the historical input sequence, $X$ (Wang et al., 2025c; Li et al., 2024b;c). A direct correlation between labels at different time steps, such as $Y_t$ and $Y_{t'}$, may not exist. However, failing to control for the common influence of $X$ can introduce spurious correlations (Wang et al., 2023a; 2025a), leading to a biased estimation (Wang et al., 2025b; Li et al., 2024a). Consequently, standard metrics like the Pearson correlation coefficient are inadequate for this task, as they are unable to isolate the relationship between $Y_t$ and $Y_{t'}$ from the spurious correlations.

To overcome this limitation, we utilize the partial correlation coefficient to provide a proxy of label autocorrelation. Our approach mirrors MATLAB's 'partialcorr' function[3]. Specifically, to compute the partial correlation between two points in the label sequence, $Y_t$ and $Y_{t'}$, while conditioning on the history sequence $X$ (the control variables), we employ a two-stage regression process. First, we fit two separate linear regression models using ordinary least squares (OLS) to predict $Y_t$ and $Y_{t'}$ from $X$. The resulting residuals, $\epsilon_t$ and $\epsilon_{t'}$, represent the variance in $Y_t$ and $Y_{t'}$ that is not explained by $X$. The partial correlation is then computed as the standard Pearson correlation between these two sets of residuals, $\rho(\epsilon_t, \epsilon_{t'})$. This procedure effectively quantifies the linear relationship between $Y_t$ and $Y_{t'}$ after factoring out the confounding influence of the historical context.

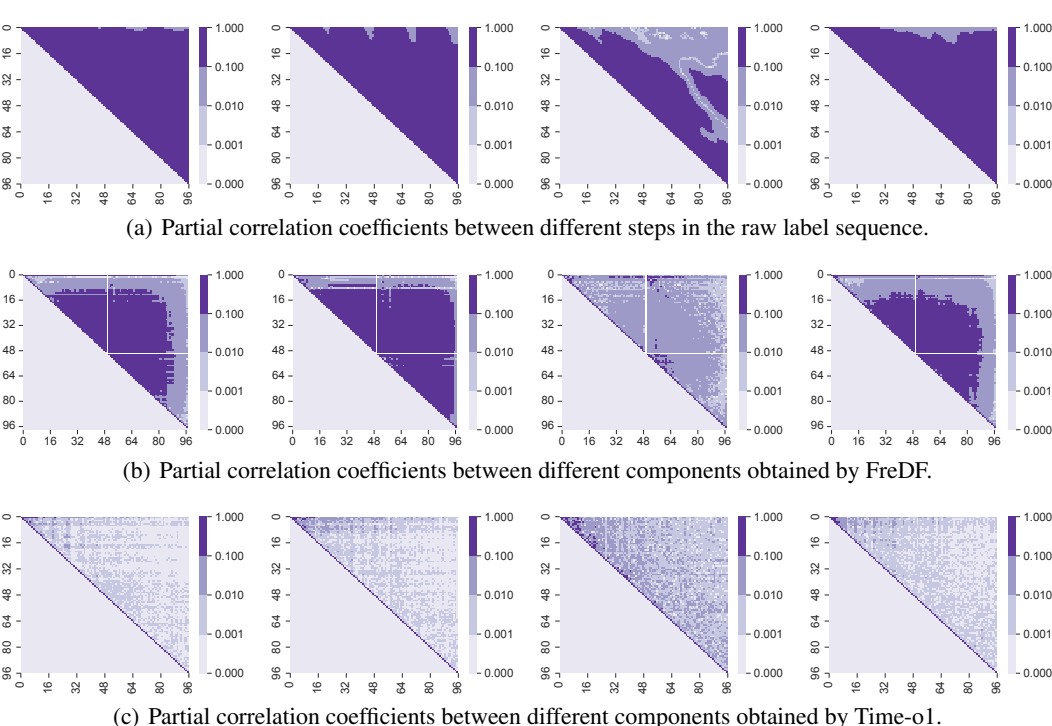

(a) Partial correlation coefficients between different steps in the raw label sequence.

(b) Partial correlation coefficients between different components obtained by FreDF.

(c) Partial correlation coefficients between different components obtained by Time-o1.

Figure 5: The label autocorrelation effect on the original label sequence and the components extracted by FreDF and Time-o1 (Wang et al., 2025d;e). The datasets are ETTh1, ETTh2, ECL, and Weather from left to right. The forecast length is uniformly set to 96.

To further validate the observations from the case study in Fig. 1, we extend the analysis on four additional datasets. As illustrated in Fig. 5, the partial correlation matrices corresponding to the raw labels display significant off-diagonal values across multiple datasets. This pattern provides strong evidence for the widespread presence of label autocorrelation. In contrast, while the latent components extracted by methods such as FreDF and Time-o1 (Wang et al., 2025e;d) show a marked reduction in these off-diagonal correlations, they do not succeed in eliminating them entirely. The persistence of

---

[3]The official implementation is detailed at https://www.mathworks.com/help/stats/partialcorr.html.

these residual values suggests that these methods only partially eliminate the autocorrelation effect. Therefore, directly applying point-wise error (such as MSE or MAE) on the obtained components yields bias due to the oversight of residual autocorrelation effect.

One might advocate for directly estimating the conditional covariance from data statistically. However, this approach is generally intractable due to its prohibitive computational complexity. Specifically, to estimate the partial correlation between each pair of time steps $t$ and $t'$, two OLS problems must be solved over the entire dataset. The scale of each OLS problem grows rapidly with the length of the history sequence and the number of covariates. Worse still, the overall complexity increases quadratically with the forecast horizon. For example, if the forecast length $\mathrm{T} = 720$, computing the full partial correlation matrix requires estimating $720 \times 720$ partial correlations. In our case study, we mitigate this complexity by subsampling only 5,000 examples from each dataset, restricting the history sequence length to 8, and limiting the forecast horizon to 96. This reduction makes the estimation tractable and affordable at the cost of accuracy, which is acceptable since the estimated results are used solely for the case study rather than for model training.

## B  THEORETICAL JUSTIFICATION

**Theorem B.1** (Likelihood formulation, Theorem 3.1 in the main text). *Given history sequence $\boldsymbol{X}$, let $\boldsymbol{Y} \in \mathbb{R}^{\mathrm{T}}$ be the associated label sequence and $g_\theta(\boldsymbol{X}) \in \mathbb{R}^{\mathrm{T}}$ be the forecast sequence. Assuming the label sequence given $\boldsymbol{X}$ follow a multivariate Gaussian distribution, the NLL of the label sequence, omitting constant terms, is:*

$$\mathcal{L}_{\boldsymbol{\Sigma}}(\boldsymbol{X}, \boldsymbol{Y}; g_\theta) = \|\boldsymbol{Y} - g_\theta(\boldsymbol{X})\|_{\bar{\boldsymbol{\Sigma}}}^2 = (\boldsymbol{Y} - g_\theta(\boldsymbol{X}))^\top \bar{\boldsymbol{\Sigma}}(\boldsymbol{Y} - g_\theta(\boldsymbol{X})), \quad (4)$$

*where $\boldsymbol{\Sigma} \in \mathbb{R}^{\mathrm{T} \times \mathrm{T}}$ is the conditional covariance of the label sequence given $\boldsymbol{X}$.*

*Proof.* The proof follows the standard derivation of negative log-likelihood given Gaussian assumption. Suppose the label sequence given $\boldsymbol{X}$ follows a multivariate normal distribution with mean vector $g_\theta(\boldsymbol{X})$ and covariance matrix $\boldsymbol{\Sigma}$. The conditional likelihood of $\boldsymbol{Y}$ is:

$$\mathbb{P}_{\boldsymbol{Y}|\boldsymbol{X}} = \frac{1}{(2\pi)^{0.5\mathrm{T}}|\boldsymbol{\Sigma}|^{0.5}} \exp(-\frac{1}{2}\|\boldsymbol{Y} - g_\theta(\boldsymbol{X})\|_{\bar{\boldsymbol{\Sigma}}}^2) \quad (5)$$

On the basis, the conditional negative log-likelihood of $\boldsymbol{Y}$ is:

$$-\log \mathbb{P}_{\boldsymbol{Y}|\boldsymbol{X}} = \frac{1}{2}\left(\mathrm{T}\log(2\pi) + \log|\boldsymbol{\Sigma}| + \|\boldsymbol{Y} - g_\theta(\boldsymbol{X})\|_{\bar{\boldsymbol{\Sigma}}}^2\right).$$

Removing the terms unrelated to $g_\theta$, the terms used for updating $\theta$ is expressed as follows:

$$\mathcal{L}_{\boldsymbol{\Sigma}}(\boldsymbol{X}, \boldsymbol{Y}; g_\theta) = \|\boldsymbol{Y} - g_\theta(\boldsymbol{X})\|_{\bar{\boldsymbol{\Sigma}}}^2. \quad (6)$$

The proof is therefore completed. □

## C  REPRODUCTION DETAILS

### C.1  DATASET DESCRIPTIONS

Our empirical evaluation is conducted on a diverse collection of widely-used time-series benchmarks, with their key properties summarized in Table 5. These include:

- **ETT** (Li et al., 2021): Electricity transformer data consisting of four subsets with varied temporal resolutions (ETTh1/ETTh2 at 1-hour intervals, ETTm1/ETTm2 at 15-minute intervals).
- **Weather** (Wu et al., 2021): Comprises 21 meteorological indicators recorded every 10 minutes from the Max Planck Institute.
- **ECL** (Wu et al., 2021): Hourly electricity consumption data from 321 clients.
- **PEMS** (Liu et al., 2022): California traffic data aggregated in 5-minute windows. We utilize the PEMS03 and PEMS08 subsets.

For all datasets, we adopt a standard chronological split into training, validation, and testing sets, following established protocols (Qiu et al., 2024; Liu et al., 2024). We standardize the input sequence length to 96 for the ETT, Weather, and ECL datasets, evaluating on forecast horizons of $\{96, 192, 336, 720\}$. For the PEMS datasets, we use forecast horizons of $\{12, 24, 36, 48\}$.

Table 5: Dataset description.

| Dataset | D | Forecast length | Train / validation / test | Frequency | Domain |
|---|---|---|---|---|---|
| ETTh1 | 7 | 96, 192, 336, 720 | 8545/2881/2881 | Hourly | Health |
| ETTh2 | 7 | 96, 192, 336, 720 | 8545/2881/2881 | Hourly | Health |
| ETTm1 | 7 | 96, 192, 336, 720 | 34465/11521/11521 | 15min | Health |
| ETTm2 | 7 | 96, 192, 336, 720 | 34465/11521/11521 | 15min | Health |
| Weather | 21 | 96, 192, 336, 720 | 36792/5271/10540 | 10min | Weather |
| ECL | 321 | 96, 192, 336, 720 | 18317/2633/5261 | Hourly | Electricity |
| PEMS03 | 358 | 12, 24, 36, 48 | 15617/5135/5135 | 5min | Transportation |
| PEMS08 | 170 | 12, 24, 36, 48 | 10690/3548/265 | 5min | Transportation |

*Note*: *D* denotes the number of variates. *Frequency* denotes the sampling interval of time points. *Train, Validation, Test* denotes the number of samples employed in each split. The taxonomy aligns with (Wu et al., 2023).

## C.2 IMPLEMENTATION DETAILS

All baseline models were reproduced using official training scripts from the iTransformer (Liu et al., 2024) and TQNet (Lin et al., 2025) repositories after checking reproducibility. Models were trained to minimize the MSE loss using the Adam optimizer (Kingma and Ba, 2015). The learning rate was selected from the set $\{10^{-3}, 5 \times 10^{-4}, 10^{-4}, 5 \times 10^{-5}\}$. We employed an early stopping patience of 3, halting training if validation loss did not improve for three consecutive epochs.

When integrating QDF into an existing forecasting model, we retained the original model's established hyperparameters as reported in public benchmarks (Liu et al., 2024; Piao et al., 2024). Our tuning was conservatively limited to the QDF-specific parameters, i.e., the round of inner update ($N_{in}$), the number of splits (K), and the update rate ($\eta$), along with the learning rate. The final hyperparameter configuration for each model was selected based on its performance on the validation set.

# D MORE EXPERIMENTAL RESULTS

## D.1 OVERALL PERFORMANCE

We provide additional experiment results of overall performance in Table 6, where the performance of each forecast horizon T is reported separately.

## D.2 SHOWCASES

We provide additional experiment results of qualitative examples in Fig. 6 and Fig. 7.

## D.3 LEARNING OBJECTIVE COMPARISON

We provide additional experiment results of learning objective comparison in Table 13.

## D.4 GENERALIZATION STUDIES

We provide additional experiment results of generalization studies in Fig. 8.

## D.5 CASE STUDY WITH PATCHTST OF VARYING HISTORICAL LENGTHS

We provide additional experiment results of varying historical lengths in Table 8, complementing the fixed length of 96 used in the main text. The forecast models selected include TQNet (Lin et al., 2025) which is the recent state-of-the-art forecast model, and PatchTST (Nie et al., 2023) which is known to require large historical lengths. The results demonstrate that QDF consistently improves both forecast models across different history sequence lengths.

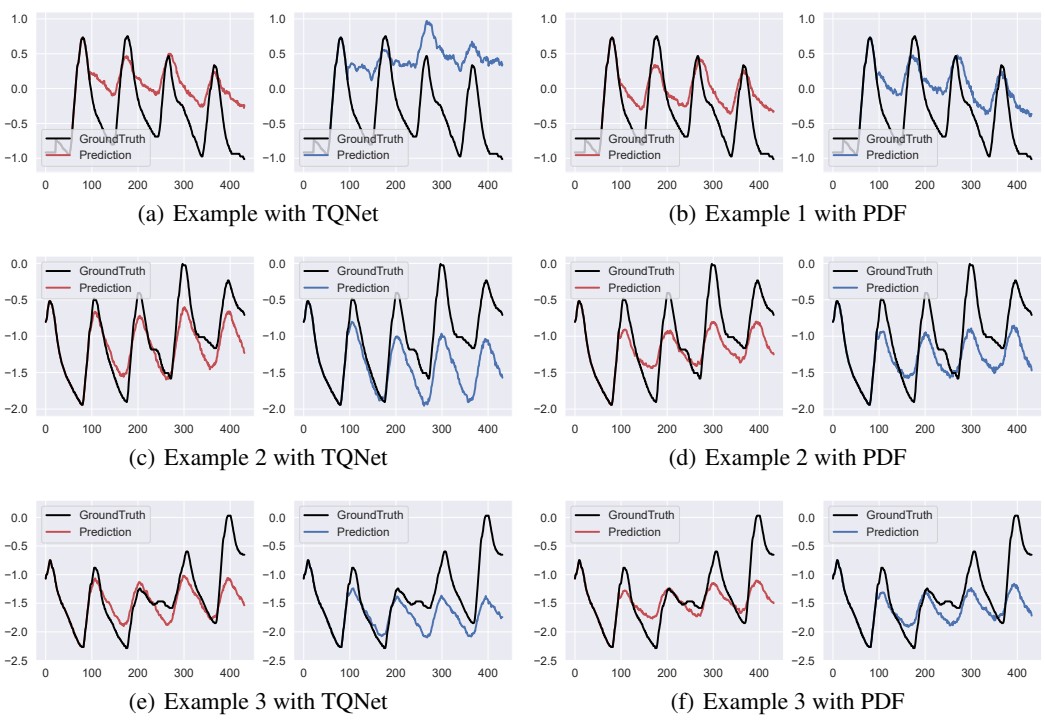

Figure 6: The forecast sequences generated with DF and QDF. The forecast length is set to 336 and the experiment is conducted on ETTm2.

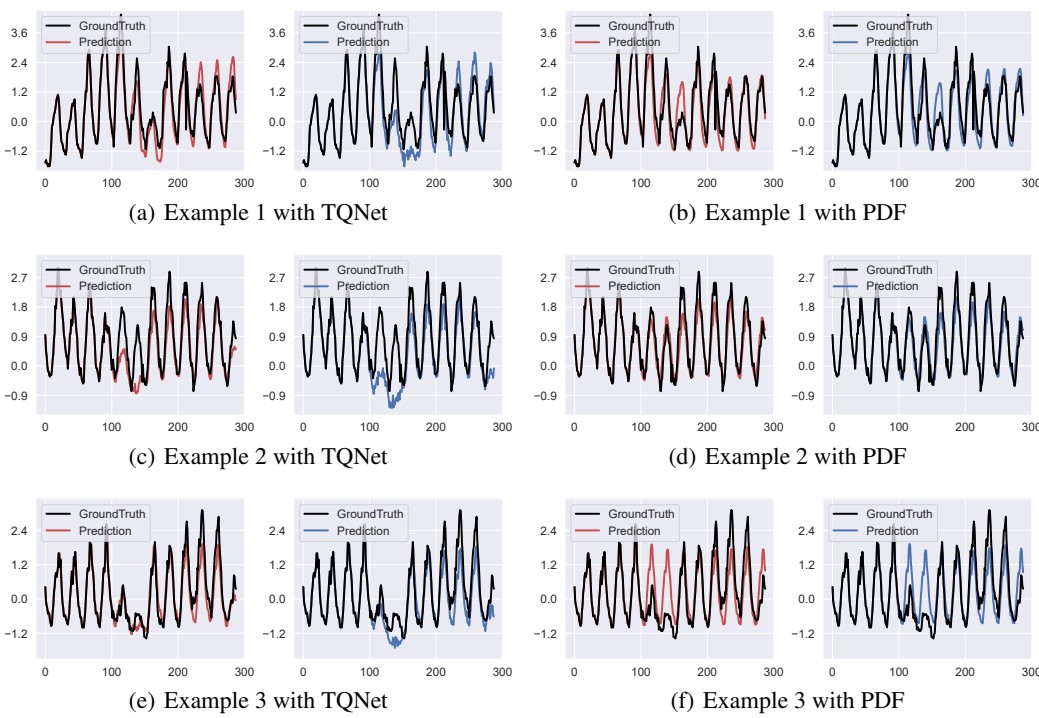

Figure 7: The forecast sequences generated with DF and QDF. The forecast length is set to 192 and the experiment is conducted on ECL.

Table 6: Full results on the multi-step forecasting task. The length of history window is set to 96 for all baselines. `Avg` indicates the results averaged over forecasting lengths: T=96, 192, 336 and 720.

| Models | | QDF (Ours) | | TQNet (2025) | | PDF (2024) | | Fredformer (2024) | | iTransformer (2024) | | FreTS (2023) | | TimesNet (2023) | | MICN (2023) | | TiDE (2023) | | PatchTST (2023) | | DLinear (2023) | |
|---|---|---|---|---|---|---|---|---|---|---|---|---|---|---|---|---|---|---|---|---|---|---|---|
| Metrics | | MSE | MAE | MSE | MAE | MSE | MAE | MSE | MAE | MSE | MAE | MSE | MAE | MSE | MAE | MSE | MAE | MSE | MAE | MSE | MAE | MSE | MAE |
| ETTm1 | 96 | 0.307 | 0.349 | 0.310 | 0.352 | 0.326 | 0.363 | 0.326 | 0.361 | 0.338 | 0.372 | 0.342 | 0.375 | 0.368 | 0.394 | 0.319 | 0.366 | 0.353 | 0.374 | 0.325 | 0.364 | 0.346 | 0.373 |
| | 192 | 0.352 | 0.376 | 0.356 | 0.377 | 0.365 | 0.381 | 0.365 | 0.382 | 0.382 | 0.396 | 0.385 | 0.400 | 0.406 | 0.409 | 0.364 | 0.395 | 0.391 | 0.393 | 0.363 | 0.383 | 0.380 | 0.390 |
| | 336 | 0.383 | 0.398 | 0.388 | 0.400 | 0.397 | 0.402 | 0.396 | 0.404 | 0.427 | 0.424 | 0.416 | 0.421 | 0.454 | 0.444 | 0.395 | 0.425 | 0.423 | 0.414 | 0.404 | 0.413 | 0.413 | 0.414 |
| | 720 | 0.441 | 0.434 | 0.450 | 0.437 | 0.458 | 0.437 | 0.459 | 0.444 | 0.496 | 0.463 | 0.513 | 0.489 | 0.527 | 0.474 | 0.505 | 0.499 | 0.486 | 0.448 | 0.463 | 0.442 | 0.472 | 0.450 |
| | Avg | 0.371 | 0.389 | 0.376 | 0.391 | 0.387 | 0.396 | 0.387 | 0.398 | 0.411 | 0.414 | 0.414 | 0.421 | 0.438 | 0.430 | 0.396 | 0.421 | 0.413 | 0.407 | 0.389 | 0.400 | 0.403 | 0.407 |
| ETTm2 | 96 | 0.170 | 0.253 | 0.175 | 0.256 | 0.176 | 0.264 | 0.177 | 0.260 | 0.182 | 0.265 | 0.188 | 0.279 | 0.184 | 0.262 | 0.178 | 0.277 | 0.182 | 0.265 | 0.180 | 0.266 | 0.188 | 0.283 |
| | 192 | 0.234 | 0.294 | 0.243 | 0.300 | 0.245 | 0.310 | 0.242 | 0.300 | 0.257 | 0.315 | 0.264 | 0.329 | 0.257 | 0.308 | 0.266 | 0.343 | 0.247 | 0.304 | 0.285 | 0.339 | 0.280 | 0.356 |
| | 336 | 0.290 | 0.331 | 0.297 | 0.336 | 0.305 | 0.345 | 0.302 | 0.340 | 0.320 | 0.354 | 0.322 | 0.369 | 0.315 | 0.345 | 0.299 | 0.354 | 0.307 | 0.343 | 0.309 | 0.347 | 0.375 | 0.420 |
| | 720 | 0.387 | 0.389 | 0.394 | 0.393 | 0.404 | 0.403 | 0.399 | 0.397 | 0.423 | 0.411 | 0.489 | 0.482 | 0.452 | 0.421 | 0.489 | 0.482 | 0.408 | 0.398 | 0.437 | 0.422 | 0.526 | 0.508 |
| | Avg | 0.270 | 0.317 | 0.277 | 0.321 | 0.283 | 0.331 | 0.280 | 0.324 | 0.295 | 0.336 | 0.316 | 0.365 | 0.302 | 0.334 | 0.308 | 0.364 | 0.286 | 0.328 | 0.303 | 0.344 | 0.342 | 0.392 |
| ETTh1 | 96 | 0.365 | 0.389 | 0.372 | 0.391 | 0.388 | 0.400 | 0.377 | 0.396 | 0.385 | 0.405 | 0.398 | 0.409 | 0.399 | 0.418 | 0.381 | 0.416 | 0.387 | 0.395 | 0.381 | 0.400 | 0.389 | 0.404 |
| | 192 | 0.427 | 0.421 | 0.430 | 0.424 | 0.440 | 0.428 | 0.437 | 0.425 | 0.440 | 0.437 | 0.451 | 0.442 | 0.452 | 0.451 | 0.497 | 0.489 | 0.439 | 0.425 | 0.450 | 0.443 | 0.442 | 0.440 |
| | 336 | 0.466 | 0.449 | 0.486 | 0.454 | 0.483 | 0.449 | 0.486 | 0.449 | 0.480 | 0.457 | 0.501 | 0.472 | 0.488 | 0.469 | 0.589 | 0.555 | 0.482 | 0.447 | 0.501 | 0.470 | 0.488 | 0.467 |
| | 720 | 0.466 | 0.467 | 0.507 | 0.486 | 0.495 | 0.482 | 0.488 | 0.467 | 0.504 | 0.492 | 0.608 | 0.571 | 0.549 | 0.515 | 0.665 | 0.617 | 0.484 | 0.471 | 0.504 | 0.492 | 0.505 | 0.502 |
| | Avg | 0.431 | 0.431 | 0.449 | 0.439 | 0.452 | 0.440 | 0.447 | 0.434 | 0.452 | 0.448 | 0.489 | 0.474 | 0.472 | 0.463 | 0.533 | 0.519 | 0.448 | 0.435 | 0.459 | 0.451 | 0.456 | 0.453 |
| ETTh2 | 96 | 0.286 | 0.338 | 0.293 | 0.343 | 0.291 | 0.340 | 0.293 | 0.344 | 0.301 | 0.349 | 0.315 | 0.374 | 0.321 | 0.358 | 0.351 | 0.398 | 0.291 | 0.340 | 0.299 | 0.349 | 0.330 | 0.383 |
| | 192 | 0.361 | 0.388 | 0.364 | 0.390 | 0.374 | 0.391 | 0.372 | 0.391 | 0.383 | 0.397 | 0.466 | 0.467 | 0.418 | 0.417 | 0.492 | 0.489 | 0.376 | 0.392 | 0.383 | 0.404 | 0.439 | 0.450 |
| | 336 | 0.408 | 0.422 | 0.411 | 0.424 | 0.414 | 0.426 | 0.420 | 0.433 | 0.425 | 0.432 | 0.522 | 0.502 | 0.464 | 0.454 | 0.656 | 0.582 | 0.417 | 0.427 | 0.439 | 0.444 | 0.589 | 0.538 |
| | 720 | 0.419 | 0.439 | 0.430 | 0.444 | 0.421 | 0.440 | 0.421 | 0.439 | 0.436 | 0.448 | 0.792 | 0.643 | 0.434 | 0.450 | 0.981 | 0.718 | 0.429 | 0.446 | 0.438 | 0.455 | 0.757 | 0.626 |
| | Avg | 0.368 | 0.397 | 0.375 | 0.400 | 0.375 | 0.399 | 0.377 | 0.402 | 0.386 | 0.407 | 0.524 | 0.496 | 0.409 | 0.420 | 0.620 | 0.546 | 0.378 | 0.401 | 0.390 | 0.413 | 0.529 | 0.499 |
| ECL | 96 | 0.135 | 0.229 | 0.143 | 0.237 | 0.175 | 0.259 | 0.161 | 0.258 | 0.150 | 0.242 | 0.180 | 0.266 | 0.170 | 0.272 | 0.170 | 0.281 | 0.197 | 0.274 | 0.170 | 0.264 | 0.197 | 0.282 |
| | 192 | 0.153 | 0.245 | 0.161 | 0.252 | 0.182 | 0.266 | 0.174 | 0.269 | 0.168 | 0.259 | 0.184 | 0.272 | 0.183 | 0.282 | 0.185 | 0.297 | 0.197 | 0.277 | 0.179 | 0.273 | 0.197 | 0.286 |
| | 336 | 0.169 | 0.262 | 0.178 | 0.270 | 0.197 | 0.282 | 0.194 | 0.290 | 0.182 | 0.274 | 0.199 | 0.290 | 0.203 | 0.302 | 0.190 | 0.298 | 0.212 | 0.292 | 0.195 | 0.288 | 0.209 | 0.301 |
| | 720 | 0.202 | 0.290 | 0.218 | 0.303 | 0.237 | 0.315 | 0.235 | 0.319 | 0.214 | 0.304 | 0.234 | 0.322 | 0.294 | 0.366 | 0.221 | 0.329 | 0.254 | 0.325 | 0.234 | 0.320 | 0.245 | 0.334 |
| | Avg | 0.165 | 0.257 | 0.175 | 0.265 | 0.198 | 0.281 | 0.191 | 0.284 | 0.179 | 0.270 | 0.199 | 0.288 | 0.212 | 0.306 | 0.192 | 0.302 | 0.215 | 0.292 | 0.195 | 0.286 | 0.212 | 0.301 |
| Weather | 96 | 0.158 | 0.201 | 0.160 | 0.203 | 0.181 | 0.221 | 0.180 | 0.220 | 0.171 | 0.210 | 0.174 | 0.228 | 0.183 | 0.229 | 0.179 | 0.244 | 0.192 | 0.232 | 0.189 | 0.230 | 0.194 | 0.253 |
| | 192 | 0.207 | 0.245 | 0.210 | 0.247 | 0.232 | 0.262 | 0.222 | 0.258 | 0.246 | 0.278 | 0.213 | 0.266 | 0.242 | 0.276 | 0.242 | 0.310 | 0.240 | 0.270 | 0.228 | 0.262 | 0.238 | 0.296 |
| | 336 | 0.263 | 0.286 | 0.267 | 0.289 | 0.285 | 0.300 | 0.283 | 0.301 | 0.296 | 0.313 | 0.270 | 0.316 | 0.293 | 0.312 | 0.273 | 0.330 | 0.292 | 0.307 | 0.288 | 0.305 | 0.282 | 0.332 |
| | 720 | 0.342 | 0.339 | 0.346 | 0.342 | 0.360 | 0.348 | 0.358 | 0.348 | 0.362 | 0.353 | 0.337 | 0.362 | 0.366 | 0.361 | 0.360 | 0.399 | 0.364 | 0.353 | 0.362 | 0.354 | 0.347 | 0.385 |
| | Avg | 0.242 | 0.268 | 0.246 | 0.270 | 0.265 | 0.283 | 0.261 | 0.282 | 0.269 | 0.289 | 0.249 | 0.293 | 0.271 | 0.295 | 0.264 | 0.321 | 0.272 | 0.291 | 0.267 | 0.288 | 0.265 | 0.317 |
| PEMS03 | 12 | 0.064 | 0.167 | 0.097 | 0.180 | 0.092 | 0.204 | 0.081 | 0.191 | 0.072 | 0.179 | 0.085 | 0.198 | 0.094 | 0.201 | 0.096 | 0.217 | 0.117 | 0.226 | 0.092 | 0.210 | 0.105 | 0.220 |
| | 24 | 0.080 | 0.189 | 0.099 | 0.204 | 0.149 | 0.261 | 0.121 | 0.240 | 0.104 | 0.217 | 0.129 | 0.244 | 0.116 | 0.221 | 0.095 | 0.210 | 0.233 | 0.322 | 0.144 | 0.263 | 0.183 | 0.297 |
| | 36 | 0.098 | 0.208 | 0.123 | 0.230 | 0.210 | 0.314 | 0.180 | 0.292 | 0.137 | 0.251 | 0.173 | 0.286 | 0.134 | 0.237 | 0.107 | 0.223 | 0.379 | 0.418 | 0.200 | 0.309 | 0.258 | 0.361 |
| | 48 | 0.112 | 0.223 | 0.157 | 0.256 | 0.275 | 0.364 | 0.201 | 0.316 | 0.174 | 0.285 | 0.207 | 0.315 | 0.161 | 0.262 | 0.125 | 0.242 | 0.535 | 0.516 | 0.245 | 0.344 | 0.319 | 0.410 |
| | Avg | 0.089 | 0.197 | 0.119 | 0.217 | 0.181 | 0.286 | 0.146 | 0.260 | 0.122 | 0.233 | 0.149 | 0.261 | 0.126 | 0.230 | 0.106 | 0.223 | 0.316 | 0.370 | 0.170 | 0.282 | 0.216 | 0.322 |
| PEMS08 | 12 | 0.074 | 0.176 | 0.079 | 0.183 | 0.100 | 0.209 | 0.091 | 0.199 | 0.084 | 0.187 | 0.096 | 0.205 | 0.111 | 0.208 | 0.161 | 0.274 | 0.121 | 0.233 | 0.106 | 0.223 | 0.113 | 0.225 |
| | 24 | 0.104 | 0.208 | 0.117 | 0.222 | 0.168 | 0.273 | 0.138 | 0.245 | 0.123 | 0.227 | 0.151 | 0.258 | 0.139 | 0.232 | 0.127 | 0.237 | 0.232 | 0.325 | 0.162 | 0.275 | 0.199 | 0.302 |
| | 36 | 0.134 | 0.237 | 0.158 | 0.260 | 0.244 | 0.333 | 0.199 | 0.303 | 0.170 | 0.268 | 0.203 | 0.303 | 0.168 | 0.260 | 0.148 | 0.252 | 0.376 | 0.427 | 0.234 | 0.331 | 0.295 | 0.371 |
| | 48 | 0.168 | 0.263 | 0.203 | 0.295 | 0.327 | 0.389 | 0.255 | 0.338 | 0.218 | 0.306 | 0.247 | 0.334 | 0.189 | 0.272 | 0.175 | 0.270 | 0.543 | 0.527 | 0.301 | 0.382 | 0.389 | 0.429 |
| | Avg | 0.120 | 0.221 | 0.139 | 0.240 | 0.210 | 0.301 | 0.171 | 0.271 | 0.149 | 0.247 | 0.174 | 0.275 | 0.152 | 0.243 | 0.153 | 0.258 | 0.318 | 0.378 | 0.201 | 0.303 | 0.249 | 0.332 |
| 1st Count | | 39 | 39 | 0 | 0 | 0 | 0 | 0 | 0 | 0 | 0 | 1 | 0 | 0 | 0 | 0 | 0 | 0 | 0 | 1 | | 0 | 0 |

## D.6 RANDOM SEED SENSITIVITY

We provide additional experiment results of random seed sensitivity in Table 9. The results include the mean and standard deviation from experiments using five different random seeds (2021, 2022, 2023, 2024, 2025) in Table 9, which showcase minimal sensitivity to random seeds.

## D.7 COMPLEXITY

We provide additional experiment results of the running time of QDF in Fig. 9. Specifically, we investigate (1) the complexity of each inner-loop update, i.e., calculating $\mathcal{L}_{\Sigma}$ with fixed $\Sigma$ for updating $\theta$, and (2) the complexity of each outer-loop update, i.e., calculating $\mathcal{L}_{\Sigma}$ with fixed $\theta$ for updating $\Sigma$. The forward phase calculates $\mathcal{L}_{\Sigma}$ while the backward phase performs updates.

As expected, the running time for both forward and backward phases increases with the forecast horizon T, since T determines the size of the weighting matrix $\Sigma$ involved in the learning objective. Nevertheless, the running time remains below 2 ms even when T increases to 720. Moreover, QDF's additional computations are confined exclusively to the training phase and are entirely isolated from inference.

Table 7: Comparable results with different learning objectives.

| Loss | | **QDF** | | Time-o1 | | FreDF | | Koopman | | Soft-DTW | | DF | |
|---|---|---|---|---|---|---|---|---|---|---|---|---|---|
| Metrics | | MSE | MAE | MSE | MAE | MSE | MAE | MSE | MAE | MSE | MAE | MSE | MAE |
| **Forecast model:TQNet** | | | | | | | | | | | | | |
| ETTm1 | 96 | 0.307 | 0.349 | 0.309 | 0.351 | 0.314 | 0.355 | 0.806 | 0.578 | 0.315 | 0.353 | 0.310 | 0.352 |
| | 192 | 0.352 | 0.376 | 0.353 | 0.375 | 0.359 | 0.378 | 0.619 | 0.515 | 0.360 | 0.377 | 0.356 | 0.377 |
| | 336 | 0.383 | 0.398 | 0.383 | 0.398 | 0.382 | 0.396 | 0.507 | 0.468 | 0.398 | 0.402 | 0.388 | 0.400 |
| | 720 | 0.441 | 0.434 | 0.444 | 0.436 | 0.444 | 0.432 | 0.450 | 0.437 | 0.476 | 0.446 | 0.450 | 0.437 |
| | Avg | 0.371 | 0.389 | 0.372 | 0.390 | 0.375 | 0.390 | 0.595 | 0.499 | 0.387 | 0.394 | 0.376 | 0.391 |
| ETTh1 | 96 | 0.365 | 0.389 | 0.381 | 0.395 | 0.369 | 0.391 | 0.415 | 0.425 | 0.379 | 0.390 | 0.372 | 0.391 |
| | 192 | 0.427 | 0.421 | 0.427 | 0.424 | 0.425 | 0.422 | 0.430 | 0.422 | 0.437 | 0.424 | 0.430 | 0.424 |
| | 336 | 0.466 | 0.449 | 0.471 | 0.444 | 0.467 | 0.445 | 0.474 | 0.445 | 0.488 | 0.453 | 0.486 | 0.454 |
| | 720 | 0.466 | 0.467 | 0.469 | 0.466 | 0.468 | 0.469 | 0.483 | 0.474 | 0.510 | 0.487 | 0.507 | 0.486 |
| | Avg | 0.431 | 0.431 | 0.437 | 0.432 | 0.432 | 0.432 | 0.451 | 0.442 | 0.453 | 0.438 | 0.449 | 0.439 |
| ECL | 96 | 0.135 | 0.229 | 0.136 | 0.228 | 0.136 | 0.228 | 0.137 | 0.231 | 0.162 | 0.258 | 0.143 | 0.237 |
| | 192 | 0.153 | 0.245 | 0.154 | 0.245 | 0.155 | 0.245 | 0.154 | 0.247 | 0.446 | 0.449 | 0.161 | 0.252 |
| | 336 | 0.169 | 0.262 | 0.171 | 0.262 | 0.172 | 0.263 | 0.171 | 0.264 | 0.912 | 0.675 | 0.178 | 0.270 |
| | 720 | 0.202 | 0.290 | 0.208 | 0.293 | 0.209 | 0.293 | 0.204 | 0.292 | 0.971 | 0.715 | 0.218 | 0.303 |
| | Avg | 0.165 | 0.257 | 0.167 | 0.257 | 0.168 | 0.257 | 0.166 | 0.258 | 0.623 | 0.524 | 0.175 | 0.265 |
| Weather | 96 | 0.158 | 0.201 | 0.159 | 0.201 | 0.158 | 0.199 | 0.223 | 0.268 | 0.161 | 0.202 | 0.160 | 0.203 |
| | 192 | 0.207 | 0.245 | 0.209 | 0.246 | 0.209 | 0.246 | 0.269 | 0.304 | 0.212 | 0.247 | 0.210 | 0.247 |
| | 336 | 0.263 | 0.286 | 0.268 | 0.290 | 0.266 | 0.288 | 0.291 | 0.309 | 0.270 | 0.289 | 0.267 | 0.289 |
| | 720 | 0.342 | 0.339 | 0.344 | 0.341 | 0.344 | 0.341 | 0.346 | 0.343 | 0.378 | 0.365 | 0.346 | 0.342 |
| | Avg | 0.242 | 0.268 | 0.245 | 0.269 | 0.244 | 0.268 | 0.282 | 0.306 | 0.255 | 0.276 | 0.246 | 0.270 |
| **Forecast model:PDF** | | | | | | | | | | | | | |
| ETTm1 | 96 | 0.320 | 0.358 | 0.326 | 0.361 | 0.325 | 0.362 | 1.051 | 0.663 | 0.323 | 0.362 | 0.326 | 0.363 |
| | 192 | 0.361 | 0.380 | 0.371 | 0.386 | 0.372 | 0.388 | 0.420 | 0.414 | 0.371 | 0.388 | 0.365 | 0.381 |
| | 336 | 0.390 | 0.401 | 0.401 | 0.409 | 0.399 | 0.409 | 0.421 | 0.415 | 0.408 | 0.413 | 0.397 | 0.402 |
| | 720 | 0.451 | 0.437 | 0.448 | 0.439 | 0.453 | 0.443 | 0.456 | 0.448 | 0.480 | 0.454 | 0.458 | 0.437 |
| | Avg | 0.381 | 0.394 | 0.386 | 0.399 | 0.387 | 0.400 | 0.587 | 0.485 | 0.396 | 0.404 | 0.387 | 0.396 |
| ETTh1 | 96 | 0.375 | 0.391 | 0.380 | 0.403 | 0.373 | 0.393 | 0.632 | 0.533 | 0.383 | 0.405 | 0.388 | 0.400 |
| | 192 | 0.423 | 0.419 | 0.422 | 0.425 | 0.423 | 0.426 | 0.424 | 0.429 | 0.430 | 0.432 | 0.440 | 0.428 |
| | 336 | 0.461 | 0.439 | 0.463 | 0.441 | 0.477 | 0.446 | 0.456 | 0.450 | 0.462 | 0.453 | 0.483 | 0.449 |
| | 720 | 0.484 | 0.468 | 0.485 | 0.483 | 0.475 | 0.476 | 0.476 | 0.478 | 0.511 | 0.496 | 0.495 | 0.482 |
| | Avg | 0.436 | 0.429 | 0.438 | 0.438 | 0.437 | 0.435 | 0.497 | 0.472 | 0.447 | 0.447 | 0.452 | 0.440 |
| ECL | 96 | 0.171 | 0.257 | 0.173 | 0.253 | 0.163 | 0.246 | 0.194 | 0.278 | 0.164 | 0.250 | 0.175 | 0.259 |
| | 192 | 0.177 | 0.261 | 0.181 | 0.262 | 0.179 | 0.261 | 0.173 | 0.260 | 0.387 | 0.410 | 0.182 | 0.266 |
| | 336 | 0.192 | 0.277 | 0.196 | 0.282 | 0.196 | 0.278 | 0.189 | 0.276 | 0.966 | 0.698 | 0.197 | 0.282 |
| | 720 | 0.234 | 0.312 | 0.229 | 0.307 | 0.237 | 0.312 | 0.228 | 0.310 | 1.263 | 0.834 | 0.237 | 0.315 |
| | Avg | 0.194 | 0.277 | 0.195 | 0.276 | 0.194 | 0.274 | 0.196 | 0.281 | 0.695 | 0.548 | 0.198 | 0.281 |
| Weather | 96 | 0.176 | 0.218 | 0.178 | 0.219 | 0.173 | 0.216 | 0.202 | 0.242 | 0.178 | 0.219 | 0.181 | 0.221 |
| | 192 | 0.225 | 0.260 | 0.236 | 0.267 | 0.235 | 0.268 | 0.225 | 0.258 | 0.232 | 0.262 | | |
| | 336 | 0.280 | 0.299 | 0.284 | 0.304 | 0.274 | 0.295 | 0.280 | 0.302 | 0.281 | 0.296 | 0.285 | 0.300 |
| | 720 | 0.357 | 0.347 | 0.357 | 0.348 | 0.356 | 0.350 | 0.353 | 0.347 | 4.502 | 1.036 | 0.360 | 0.348 |
| | Avg | 0.259 | 0.281 | 0.264 | 0.284 | 0.268 | 0.287 | 0.268 | 0.290 | 1.296 | 0.452 | 0.265 | 0.283 |

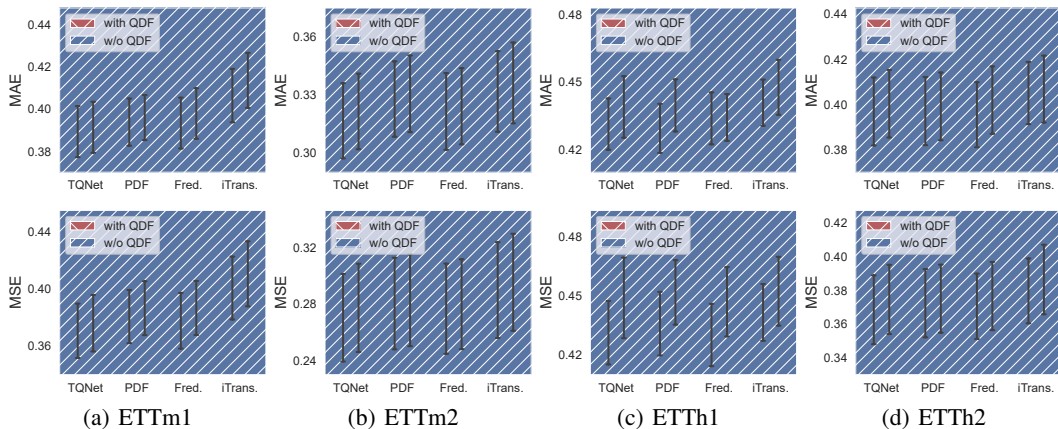

Figure 8: Performance of different forecast models with and without QDF. The forecast errors are averaged over forecast lengths and the error bars represent 50% confidence intervals.

Table 8: Varying input sequence length results on the Weather dataset.

| | Models | | QDF | | TQNet | | QDF | | PatchTST | |
|---|---|---|---|---|---|---|---|---|---|---|
| | Metrics | | MSE | MAE | MSE | MAE | MSE | MAE | MSE | MAE |
| Input sequence length | 96 | 96 | 0.158 | 0.201 | 0.160 | 0.203 | 0.180 | 0.224 | 0.189 | 0.230 |
| | | 192 | 0.207 | 0.245 | 0.210 | 0.247 | 0.226 | 0.262 | 0.228 | 0.262 |
| | | 336 | 0.263 | 0.286 | 0.267 | 0.289 | 0.279 | 0.300 | 0.288 | 0.305 |
| | | 720 | 0.342 | 0.339 | 0.346 | 0.342 | 0.354 | 0.347 | 0.362 | 0.354 |
| | | Avg | 0.242 | 0.268 | 0.246 | 0.270 | 0.260 | 0.283 | 0.267 | 0.288 |
| | 192 | 96 | 0.152 | 0.199 | 0.151 | 0.197 | 0.161 | 0.208 | 0.163 | 0.209 |
| | | 192 | 0.198 | 0.241 | 0.198 | 0.241 | 0.207 | 0.248 | 0.207 | 0.249 |
| | | 336 | 0.252 | 0.282 | 0.253 | 0.283 | 0.259 | 0.287 | 0.268 | 0.293 |
| | | 720 | 0.324 | 0.332 | 0.327 | 0.334 | 0.334 | 0.337 | 0.511 | 0.451 |
| | | Avg | 0.231 | 0.263 | 0.232 | 0.264 | 0.240 | 0.270 | 0.287 | 0.301 |
| | 336 | 96 | 0.148 | 0.198 | 0.149 | 0.198 | 0.160 | 0.214 | 0.158 | 0.208 |
| | | 192 | 0.195 | 0.240 | 0.196 | 0.243 | 0.204 | 0.253 | 0.235 | 0.291 |
| | | 336 | 0.244 | 0.279 | 0.246 | 0.281 | 0.251 | 0.287 | 0.252 | 0.287 |
| | | 720 | 0.313 | 0.327 | 0.318 | 0.331 | 0.324 | 0.338 | 0.326 | 0.336 |
| | | Avg | 0.225 | 0.261 | 0.227 | 0.263 | 0.235 | 0.273 | 0.243 | 0.280 |
| | 720 | 96 | 0.148 | 0.199 | 0.155 | 0.206 | 0.161 | 0.217 | 0.153 | 0.205 |
| | | 192 | 0.192 | 0.241 | 0.203 | 0.251 | 0.205 | 0.255 | 0.205 | 0.254 |
| | | 336 | 0.246 | 0.285 | 0.257 | 0.295 | 0.254 | 0.293 | 0.248 | 0.288 |
| | | 720 | 0.310 | 0.329 | 0.319 | 0.339 | 0.315 | 0.337 | 0.317 | 0.339 |
| | | Avg | 0.224 | 0.264 | 0.233 | 0.273 | 0.234 | 0.276 | 0.231 | 0.272 |

Table 9: Experimental results ($\mathrm{mean}_{\pm\mathrm{std}}$) with varying seeds (2021-2025).

| Dataset | | | ECL | | | | | Weather | | |
|---|---|---|---|---|---|---|---|---|---|---|
| Models | | QDF | | DF | | | QDF | | DF | |
| Metrics | | MSE | MAE | MSE | MAE | | MSE | MAE | MSE | MAE |
| 96 | | $0.135_{\pm0.000}$ | $0.229_{\pm0.000}$ | $0.143_{\pm0.000}$ | $0.237_{\pm0.000}$ | | $0.160_{\pm0.001}$ | $0.203_{\pm0.001}$ | $0.160_{\pm0.001}$ | $0.203_{\pm0.001}$ |
| 192 | | $0.153_{\pm0.000}$ | $0.245_{\pm0.000}$ | $0.161_{\pm0.000}$ | $0.252_{\pm0.000}$ | | $0.208_{\pm0.001}$ | $0.246_{\pm0.001}$ | $0.211_{\pm0.001}$ | $0.248_{\pm0.001}$ |
| 336 | | $0.169_{\pm0.000}$ | $0.262_{\pm0.000}$ | $0.178_{\pm0.000}$ | $0.270_{\pm0.000}$ | | $0.264_{\pm0.001}$ | $0.287_{\pm0.001}$ | $0.266_{\pm0.001}$ | $0.289_{\pm0.001}$ |
| 720 | | $0.202_{\pm0.002}$ | $0.291_{\pm0.002}$ | $0.218_{\pm0.000}$ | $0.303_{\pm0.000}$ | | $0.343_{\pm0.001}$ | $0.340_{\pm0.001}$ | $0.345_{\pm0.001}$ | $0.342_{\pm0.000}$ |
| Avg | | $0.165_{\pm0.001}$ | $0.257_{\pm0.000}$ | $0.175_{\pm0.000}$ | $0.265_{\pm0.000}$ | | $0.244_{\pm0.001}$ | $0.269_{\pm0.001}$ | $0.246_{\pm0.001}$ | $0.271_{\pm0.001}$ |

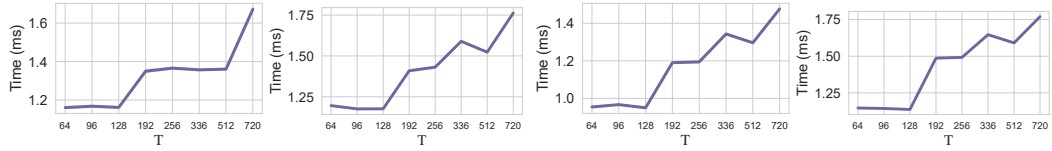

(a) Running time in the forward phase.       (b) Running time in the backward phase.

Figure 9: The running time of the QDF algorithm given varying forecast horizons (T). In each sub-figure, the left panel considers the complexity of each inner-loop update (i.e., step 4 in Algorithm 1), the right panel considers the complexity of each outer-loop update (i.e., step 7 in Algorithm 1).

Table 10: The comprehensive results of different learning objectives on the MAPE metric.

| Dataset | QDF | Time-o1 | FreDF | Koopman | Dilate | Soft-DTW | DF |
|---|---|---|---|---|---|---|---|
| **Forecast model:TQNet** | | | | | | | |
| ETTm1 | 2.305 | 2.315 | 2.313 | 2.783 | 2.338 | 2.319 | 2.338 |
| ETTh1 | 9.619 | 9.697 | 9.875 | 10.488 | 10.036 | 10.283 | 10.290 |
| ECL | 2.509 | 2.540 | 2.534 | 2.601 | 2.554 | 5.316 | 2.578 |
| Weather | 3.054 | 3.098 | 3.086 | 3.573 | 3.121 | 3.276 | 3.121 |
| **Forecast model:PDF** | | | | | | | |
| ETTm1 | 2.232 | 2.420 | 2.412 | 2.876 | 2.421 | 2.391 | 2.283 |
| ETTh1 | 9.565 | 10.563 | 10.468 | 11.255 | 10.648 | 10.980 | 9.846 |
| ECL | 2.750 | 2.795 | 2.787 | 2.757 | 2.749 | 5.199 | 2.869 |
| Weather | 3.156 | 3.226 | 3.189 | 3.293 | 3.244 | 5.064 | 3.228 |

*Therefore, QDF introduces no additional complexity to model inference, and the extra computational cost during training is minimal.*

### D.8 PERFORMANCE ON THE MAPE METRIC

We provide additional experimental results of the MAPE metric in Table 10 on four datasets. The underlying forecasting model is selected as TQNet (Lin et al., 2025) and PDF (Dai et al., 2024) for their competitive performance. Overall, QDF achieves the best performance on 7 out of 8 cases, thereby further substantiating its effectiveness on the MAPE metric.

### D.9 SHORT-TERM FORECASTING RESULT

We provide additional experimental results of the short-term forecasting task on the M4 dataset in Table 11. The forecasting architectures are selected as TQNet (Lin et al., 2025) and PDF (Dai et al., 2024) for their recency and competitive performance. Overall, QDF consistently performs best in 21/30 cases, yielding the best overall performance.

### D.10 CONVERGENCE OF ALGORITHMS

We visualize the loss curves of Algorithm 1 and 2 in Fig. 10 to demonstrate the convergence of the two algorithms.

### D.11 VISUALIZATION OF THE CORRELATION MATRIX

We provide additional experimental results to visualize the correlation matrix learned by QDF in Fig. 11.

## E STATEMENT ON THE USE OF LARGE LANGUAGE MODELS (LLMS)

In accordance with the conference guidelines, we disclose our use of Large Language Models (LLMs) in the preparation of this paper as follows:

Table 11: The comprehensive results on the short-term forecasting task.

| Loss | QDF | | | Time-o1 | | | FreDF | | | Koopman | | | DF | | |
|---|---|---|---|---|---|---|---|---|---|---|---|---|---|---|---|
| Metric | SMAPE | MASE | OWA | SMAPE | MASE | OWA | SMAPE | MASE | OWA | SMAPE | MASE | OWA | SMAPE | MASE | OWA |
| **Forecast model:TQNet** | | | | | | | | | | | | | | | |
| Yearly | 13.355 | 3.015 | 0.788 | 13.377 | 3.004 | 0.787 | 13.404 | 3.022 | 0.790 | 22.588 | 5.512 | 1.385 | 13.502 | 3.074 | 0.800 |
| Quarterly | 10.018 | 1.174 | 0.883 | 10.174 | 1.200 | 0.899 | 10.116 | 1.196 | 0.895 | 17.713 | 2.415 | 1.685 | 10.132 | 1.192 | 0.895 |
| Monthly | 12.756 | 0.939 | 0.884 | 12.776 | 0.949 | 0.889 | 12.786 | 0.952 | 0.891 | 18.655 | 1.506 | 1.355 | 12.777 | 0.945 | 0.887 |
| Others | 4.909 | 3.203 | 1.022 | 5.039 | 3.285 | 1.048 | 4.908 | 3.219 | 1.024 | 7.478 | 5.365 | 1.633 | 5.048 | 3.292 | 1.050 |
| Average | 11.844 | 1.586 | 0.851 | 11.903 | 1.599 | 0.857 | 11.894 | 1.600 | 0.857 | 18.775 | 2.839 | 1.434 | 11.923 | 1.611 | 0.861 |
| **Forecast model:PDF** | | | | | | | | | | | | | | | |
| Yearly | 13.426 | 3.044 | 0.794 | 13.524 | 3.014 | 0.793 | 13.479 | 3.052 | 0.796 | 23.515 | 5.695 | 1.436 | 13.532 | 3.036 | 0.796 |
| Quarterly | 10.361 | 1.224 | 0.917 | 10.690 | 1.282 | 0.953 | 10.367 | 1.241 | 0.923 | 19.090 | 2.572 | 1.804 | 10.646 | 1.279 | 0.950 |
| Monthly | 12.930 | 0.961 | 0.900 | 13.181 | 1.003 | 0.928 | 13.023 | 0.987 | 0.916 | 20.595 | 1.756 | 1.540 | 13.208 | 0.999 | 0.928 |
| Others | 4.891 | 3.262 | 1.029 | 5.012 | 3.256 | 1.041 | 5.381 | 3.579 | 1.130 | 9.890 | 8.213 | 2.336 | 5.698 | 3.735 | 1.188 |
| Average | 12.026 | 1.618 | 0.866 | 12.254 | 1.645 | 0.882 | 12.108 | 1.653 | 0.879 | 20.370 | 3.181 | 1.583 | 12.292 | 1.672 | 0.890 |

(a) The dynamics of loss functions in Algorithm 1 across different epochs.

(b) The dynamics of loss functions in Algorithm 2 across different epochs.

Figure 10: The dynamics of loss functions on ETTm1, ETTm2, ETTh1, ETTh2, ECL, and Weather from left to right. The forecast horizon $T = 336$.

Table 12: Running complexity of QDF algorithm under task splits K=3.

| Direction | Loop | T=64 | T=96 | T=128 | T=192 | T=256 | T=336 | T=512 | T=720 |
|---|---|---|---|---|---|---|---|---|---|
| Forward | Inner | $1.196_{\pm 0.007}$ | $1.175_{\pm 0.011}$ | $1.176_{\pm 0.009}$ | $1.409_{\pm 0.012}$ | $1.431_{\pm 0.015}$ | $1.590_{\pm 0.011}$ | $1.523_{\pm 0.012}$ | $1.763_{\pm 0.011}$ |
| | Outer | $1.161_{\pm 0.010}$ | $1.168_{\pm 0.014}$ | $1.162_{\pm 0.009}$ | $1.350_{\pm 0.019}$ | $1.366_{\pm 0.011}$ | $1.357_{\pm 0.013}$ | $1.361_{\pm 0.012}$ | $1.672_{\pm 0.012}$ |
| Backward | Inner | $1.147_{\pm 0.005}$ | $1.144_{\pm 0.005}$ | $1.137_{\pm 0.006}$ | $1.487_{\pm 0.007}$ | $1.492_{\pm 0.008}$ | $1.647_{\pm 0.007}$ | $1.591_{\pm 0.010}$ | $1.770_{\pm 0.010}$ |
| | Outer | $0.954_{\pm 0.006}$ | $0.967_{\pm 0.006}$ | $0.950_{\pm 0.008}$ | $1.190_{\pm 0.009}$ | $1.194_{\pm 0.009}$ | $1.343_{\pm 0.008}$ | $1.296_{\pm 0.007}$ | $1.477_{\pm 0.008}$ |

Table 13: The performance of varying ranks of the inverse covariance matrix $\bar{\Sigma}$.

| Rank | | Rank=1 | | | Rank=0.8 | | | Rank=0.6 | | | Rank=0.4 | | | Rank=0.2 | | | DF | |
|---|---|---|---|---|---|---|---|---|---|---|---|---|---|---|---|---|---|---|
| Metrics | MSE | MAE | MAPE | MSE | MAE | MAPE | MSE | MAE | MAPE | MSE | MAE | MAPE | MSE | MAE | MAPE | MSE | MAE | MAPE |
| **Forecast model:TQNet** | | | | | | | | | | | | | | | | | | |
| ETTm1 96 | 0.307 | 0.349 | 2.156 | 0.308 | 0.350 | 2.191 | 0.310 | 0.351 | 2.174 | 0.309 | 0.351 | 2.200 | 0.310 | 0.352 | 2.201 | 0.310 | 0.352 | 2.212 |
| ETTm1 192 | 0.352 | 0.376 | 2.281 | 0.354 | 0.377 | 2.288 | 0.355 | 0.379 | 2.308 | 0.354 | 0.378 | 2.298 | 0.352 | 0.377 | 2.295 | 0.356 | 0.377 | 2.288 |
| ETTm1 336 | 0.383 | 0.398 | 2.329 | 0.387 | 0.399 | 2.349 | 0.387 | 0.399 | 2.344 | 0.387 | 0.400 | 2.356 | 0.387 | 0.401 | 2.357 | 0.388 | 0.400 | 2.338 |
| ETTm1 720 | 0.441 | 0.434 | 2.522 | 0.445 | 0.437 | 2.534 | 0.446 | 0.437 | 2.546 | 0.443 | 0.437 | 2.530 | 0.445 | 0.437 | 2.538 | 0.450 | 0.437 | 2.516 |
| Avg | 0.371 | 0.389 | 2.322 | 0.373 | 0.391 | 2.341 | 0.375 | 0.392 | 2.343 | 0.373 | 0.391 | 2.346 | 0.373 | 0.392 | 2.348 | 0.376 | 0.391 | 2.338 |
| Weather 96 | 0.158 | 0.201 | 2.619 | 0.158 | 0.201 | 2.621 | 0.159 | 0.202 | 2.635 | 0.158 | 0.201 | 2.653 | 0.159 | 0.202 | 2.602 | 0.160 | 0.203 | 2.642 |
| Weather 192 | 0.207 | 0.245 | 3.047 | 0.208 | 0.246 | 3.035 | 0.208 | 0.246 | 3.051 | 0.207 | 0.246 | 3.049 | 0.207 | 0.246 | 3.065 | 0.210 | 0.247 | 3.087 |
| Weather 336 | 0.263 | 0.286 | 3.302 | 0.264 | 0.287 | 3.343 | 0.264 | 0.287 | 3.330 | 0.263 | 0.287 | 3.323 | 0.264 | 0.287 | 3.337 | 0.267 | 0.289 | 3.352 |
| Weather 720 | 0.342 | 0.339 | 3.372 | 0.342 | 0.339 | 3.354 | 0.342 | 0.339 | 3.378 | 0.342 | 0.339 | 3.355 | 0.342 | 0.340 | 3.380 | 0.346 | 0.342 | 3.403 |
| Avg | 0.242 | 0.268 | 3.085 | 0.243 | 0.268 | 3.088 | 0.243 | 0.269 | 3.098 | 0.243 | 0.268 | 3.095 | 0.243 | 0.269 | 3.096 | 0.246 | 0.270 | 3.121 |
| **Forecast model:PDF** | | | | | | | | | | | | | | | | | | |
| ETTm1 96 | 0.320 | 0.358 | 2.176 | 0.314 | 0.355 | 2.142 | 0.315 | 0.353 | 2.066 | 0.317 | 0.358 | 2.103 | 0.314 | 0.356 | 2.126 | 0.326 | 0.363 | 2.179 |
| ETTm1 192 | 0.361 | 0.380 | 2.210 | 0.358 | 0.378 | 2.249 | 0.358 | 0.376 | 2.191 | 0.359 | 0.379 | 2.194 | 0.361 | 0.380 | 2.253 | 0.365 | 0.381 | 2.209 |
| ETTm1 336 | 0.390 | 0.401 | 2.277 | 0.389 | 0.401 | 2.253 | 0.389 | 0.402 | 2.276 | 0.389 | 0.399 | 2.245 | 0.389 | 0.398 | 2.280 | 0.397 | 0.402 | 2.279 |
| ETTm1 720 | 0.451 | 0.437 | 2.487 | 0.449 | 0.434 | 2.469 | 0.447 | 0.434 | 2.467 | 0.448 | 0.435 | 2.468 | 0.447 | 0.435 | 2.475 | 0.458 | 0.437 | 2.467 |
| Avg | 0.381 | 0.394 | 2.287 | 0.377 | 0.392 | 2.278 | 0.377 | 0.391 | 2.250 | 0.378 | 0.393 | 2.253 | 0.378 | 0.392 | 2.283 | 0.387 | 0.396 | 2.283 |
| Weather 96 | 0.176 | 0.218 | 2.708 | 0.178 | 0.219 | 2.762 | 0.181 | 0.221 | 2.805 | 0.179 | 0.220 | 2.776 | 0.179 | 0.220 | 2.826 | 0.181 | 0.221 | 2.812 |
| Weather 192 | 0.225 | 0.260 | 3.164 | 0.226 | 0.260 | 3.142 | 0.227 | 0.261 | 3.172 | 0.226 | 0.260 | 3.165 | 0.227 | 0.260 | 3.160 | 0.232 | 0.262 | 3.213 |
| Weather 336 | 0.280 | 0.299 | 3.391 | 0.280 | 0.298 | 3.408 | 0.281 | 0.298 | 3.387 | 0.279 | 0.299 | 3.407 | 0.280 | 0.299 | 3.408 | 0.285 | 0.300 | 3.407 |
| Weather 720 | 0.357 | 0.347 | 3.443 | 0.357 | 0.347 | 3.445 | 0.357 | 0.347 | 3.457 | 0.357 | 0.348 | 3.472 | 0.356 | 0.347 | 3.440 | 0.360 | 0.348 | 3.482 |
| Avg | 0.259 | 0.281 | 3.176 | 0.260 | 0.281 | 3.189 | 0.261 | 0.282 | 3.205 | 0.260 | 0.282 | 3.205 | 0.260 | 0.282 | 3.208 | 0.265 | 0.283 | 3.228 |

(a) The learned correlation matrix given forecast horizon T = 336

(b) The learned correlation matrix given forecast horizon T = 720

Figure 11: The learned correlation matrix on different datasets: ETTm1, ETTm2 and Weather from left to right.

We used LLMs (specifically, OpenAI GPT-4.1, GPT-5 and Google Gemini 2.5) *solely for checking grammar errors and improving the readability of the manuscript*. The LLMs *were not involved in research ideation, the development of research contributions, experiment design, data analysis, or interpretation of results*. All substantive content and scientific claims were created entirely by the authors. The authors have reviewed all LLM-assisted text to ensure accuracy and originality, and take full responsibility for the contents of the paper. The LLMs are not listed as an author.

