# OpenReview forum: "Quadratic Direct Forecast for Training Multi-Step Time-Series Forecast Models"
_ICLR.cc/2026/Conference — ICLR 2026 Poster_

### Official Review · Reviewer_wfuZ · 2025-10-31

**Soundness:** 3
**Presentation:** 3
**Contribution:** 3
**Rating:** 6
**Confidence:** 4

**Summary:**

This paper investigates the learning objective design of time-series forecasting models. Specifically, it proposes a novel quadratic-form weighted training objective (QDF) to address two issues of existing methods: (1) oversight of the label autocorrelation effect among future steps, and (2) failure to set heterogeneous task weights for different forecasting tasks.  Experiments showcase the validity of QDF on public datasets.

**Strengths:**

1. The paper improves the learning objective for TSF models, which is an insufficiently explored yet important research problem in time-series forecasting.

2. Experiment results are effectively presented and structured to support the paper's claims.

3. Code is provided which facilitates reproduction.

**Weaknesses:**

1. The paper could benefit from a more analytical exploration why the proposed weighting matrix could possibly improve existing formulations, especially FreDF and Time-o1.

2. For time-series datasets, careful design of data splitting strategies is essential to avoid information leakage. It is not very clear whether the risk of leakage is fully bypassed in the hybrid splitting strategy of this paper (e.g., validation, meta-update, etc.). A detailed analysis is needed to analyze the leakage issue as well as the strategies or designs to avoid it.

3. Although the code is released, the  introduction to key components is lacking which impedes verification: how to reproduce the results with the used hyper parameters? where is the key code components? and what is the required software environment for reproducing? whether the code is built upon established repos?

**Questions:**

1. In Definition 3.2, the optimization problem optimizes $\theta$ in $D_{in}$ and then $\Sigma$ in $D_{out}$. Compared to using the full dataset, i.e., the concatenation of $D_{in}$ and $D_{out}$ to update $\Sigma$ and $\theta$, what is the advantage of the splitting strategy here?

2. Whether the proposed methodology is directly applicable to general multitask learning tasks? Is there any limitation or necessary adaptations when generalizing beyond the TSF context?

---

> ### Author Response · Authors · 2025-11-29
>
> Thank you very much for your positive comments and appreciation of our **novelty, experiments and reproducibility**. Below are our responses to the specific query raised.
>
> -------
>
> #### **[W1] The paper could benefit from a more analytical exploration why the proposed weighting matrix could possibly improve existing formulations, especially FreDF and Time-o1.**
> **Response.** Thank you for your actionable suggestion. Here, we first derive a general loss formulation for FreDF, Time-o1 and QDF, and then perform analytical analysis on the advantage of QDF. The detailed analysis is as follows.
> - Firstly, we construct a generalized learning objective $\mathcal{L}_\mathrm{gene} = \left\|\mathbf{T} \mathbf{e}\right\|_p^p$ for subsequent analytical exploration, where $\mathbf{e}=\mathbf{y}-\hat{\mathbf{y}}$ is the forecasting residual, and $\mathbf{T}$ is the linear transform matrix.
> - **Secondly, we demonstrate that FreDF, Time-o1, and QDF are all special cases of $\mathcal{L}_\mathrm{gene}$.** For FreDF, $\mathbf{T}$ is the Fourier matrix (see Definition C.4 in [1]). For Time-o1, $\mathbf{T}$ is the PCA projection matrix (see Lemma 3.3 in [2]). For QDF, $\mathbf{T}$ is the square root of the inverse correlation matrix, i.e., $\mathbf{T} =\bar{\boldsymbol{\Sigma}}^{-1/2}$, with $p=2$.
>
> - **On this basis, we can attribute QDF's advantage over FreDF and Time-o1 to the transform matrix $\mathbf{T}$.** In both FreDF and Time-o1, $\mathbf{T}$ is predetermined and remains fixed throughout training. Contrastingly, in QDF, $\mathbf{T}$ is explicitly learned from data throughout training. This learning-based approach affords QDF greater modeling flexibility and adaptiveness to dataset-specific characteristics, thereby improving the forecasting performance.
>
> [1] Optimal transport for time-series imputation. ICLR 2025.
>
> [2] Time-o1: Time-series forecasting needs transformed label alignment. NeurIPS 2025.
>
>
>
> #### **[W2] For time-series datasets, careful design of data splitting strategies is essential to avoid information leakage. It is not very clear whether the risk of leakage is fully bypassed in the hybrid splitting strategy of this paper (e.g., validation, meta-update, etc.). A detailed analysis is needed to analyze the leakage issue as well as the strategies or designs to avoid it.**
>
> **Response.** Thank you for your meticulous comment. **We would like to emphasize, with clarity and confidence, that our experimental pipeline rigorously precludes any risk of data leakage.** The evidence is straightforward: the full training pipeline (Algorithm 2) exclusively utilizes the training set, without using the validation or test sets.
>
> Some might argue that splitting the training and validation sets might introduce data leakage. Nevertheless, we mandate a strict protocol: we first split the observations into training and validation sets, and then generate the history/label sequences by sliding window. **This further ensures that there is no overlap between the training and validation sets, thus preventing data leakage.**
>
>
> #### **[W3] Although the code is released, the introduction to key components is lacking which impedes verification: how to reproduce the results with the used hyper parameters? where is the key code components? and what is the required software environment for reproducing? whether the code is built upon established repos?**
>
> **Response.** Once again, thank you very much for your actionable comment. We have revised the repository, especially the README file and the environment file, to make it easy to follow.
> - On the query `how to reproduce the results with the used hyper parameters`, we add a `Reproduction` section in the README file, where we list the scripts to reproduce the results on different datasets.
> - On the query `where is the key code components?`, we add a `Components` section in the README file, where we list the key code components and their locations.
> - On the query `what is the required software environment for reproducing?`, we add an `Environment` section in the README file, where we provide step-by-step instructions to install the required Python environment.
> - On the query `whether the code is built upon established repos?`, we add an `Acknowledgments` section in the README file, where we acknowledge related repositories.

---

> ### Author Response · Authors · 2025-11-29
>
> #### **[Q1] In Definition 3.2, the optimization problem optimizes $\theta$ in $D_{in}$ and then $\Sigma$ in $D_{out}$. Compared to using the full dataset, i.e., the concatenation of $D_{in}$ and $D_{out}$ to update $\theta$ and $\Sigma$, what is the advantage of the splitting strategy here?**
>
> **Response.** Thank you for your insightful question. For clarity, we denote `optimizing $\theta$ in $D_{in}$ and then $\Sigma$ in $D_{out}$` as the splitting strategy, and `using the full dataset to update $\theta$ and $\Sigma$ simultaneously` as the full strategy. We clarify the advantage of the splitting strategy as follows.
> - **Firstly, we note that the full strategy is susceptible to overfitting.** Specifically, this approach involves training the model parameters $\theta$ on the entire training set $D_{full}$, and subsequently tuning the loss weighting matrix $\Sigma$ on the same data. Such practice induces a significant risk of overfitting, as both the model and the loss function are optimized using identical data, potentially leading to a loss weighting matrix that merely facilitates minimizing the loss on $D_{full}$ without promoting genuine generalization.
> - **Secondly, we note that the splitting strategy can reduce the overfitting risk by data
> splitting.** Specifically, it first trains the model parameters $\theta$ using the subset $D_{in}$, followed by the optimization of the loss weighting matrix $\Sigma$ on the new subset $D_{out}$. Therefore, $\Sigma$ is trained to enhance the generalization capability of the model on previously unseen data ($D_{out}$), rather than merely improving in-sample performance, therefore less prone to overfitting. **Indeed, similar splitting strategy is widespread in meta-learning, where meta-training and meta-validation are performed separately.**
>
>
> #### **[Q2] Whether the proposed methodology is directly applicable to general multitask learning tasks? Is there any limitation or necessary adaptations when generalizing beyond the TSF context?**
> **Response.** We thank the reviewer for this thoughtful and meticulous question. We agree that
> QDF can be applied in other multitask learning scenarios, but it needs some necessary adaptations in some cases. We provide clarification on the two aspects as follows.
> - **Firstly, we agree that QDF can be applied in other multitask learning scenarios.** In this work, two challenges are addressed by QDF: label autocorrelation and heterogeneous task weighting. The two challenges are not limited to time-series forecasting, but can be significant in other multitask learning scenarios. For example, in recommendation systems, the click-through rate and the purchase rate are two correlated tasks with different weights of significance. Therefore, QDF can be applied to learn the weighting matrix for the two tasks and improve the overall recommendation performance.
> - **Secondly, it is important to emphasize that QDF is inherently formulated for multitask regression problems.** As established in Theorem 3.1, QDF is predicated on the assumption that the negative log-likelihood follows a Gaussian distribution, a standard paradigm in regression settings. This assumption does not directly extend to classification tasks. Consequently, to generalize QDF to classification scenarios, it is necessary to reformulate the negative log-likelihood—such as adopting the cross-entropy function or other suitable alternatives—to align with the characteristics of classification objectives.

---

### Official Review · Reviewer_VKq3 · 2025-10-31

**Soundness:** 2
**Presentation:** 2
**Contribution:** 3
**Rating:** 6
**Confidence:** 3

**Summary:**

This paper presents QDF, a method for training forecasting models using a trained covariance matrix, thus solving the problem of assuming uncorrelated residuals. The authors formulate learning $\Sigma$ as a bilevel problem, which they solve sequentially over $K$ splits of the data, then train the forecasting model using the learned (static) $\Sigma$. They empirically compare the QDF objective with other time series losses, explore design choices for learning the covariance and comment on the sensitivity of the introduced hyperparameters.

**Strengths:**

- The paper introduces a technically novel strategy to solve an empirically and theoretically motivated problem that is much relevant in neural, (direct) non-autocorrelational models for time series forecasting.
- Promising empirical evidence of QDF on a comprehensive set of datasets, against transformer and non-transformer based methods, and against a variety of time series optimization objectives.
- Ablation studies and the effect on various architectures is provided.

**Weaknesses:**

- A theoretical result or comments on the convergence of Algorithms 1 and 2 would make the presentation stronger.
- Emphasizing the above point, the proposed method may be too computationally expensive for some models, given the bilevel optimization, over $K$ splits required to perform the optimization to find $\Sigma$.
- Some parts of the text are not very clear: For instance, on the results in Section 4.2, which model is used to compare the different forecasting objectives? Similarly in Table 3 (Ablation study).
- The clarity  of the last paragraph of Secton 3.2 could be improved.
- The connection of this method with meta-learning is not very clear as it is not formally stated.
- There's the unmentioned assumption that the time series is non-heteroscedastic and thus is characterized by a single $\Sigma$

**Questions:**

- Are some architectures more prone to converge to correlated residuals than others?
- In the Ablation  study (4.4) how are the two QDFs integrated, by taking their average?
- Visualizing the correlation matrices of the residuals after training with QDF would be interesting to see.

---

> ### Author Response · Authors · 2025-11-29
>
> We would like to express our sincere gratitude for your positive evaluation and appreciation of our **novelty, theoretical motivation, performance and ablation studies**. We would like to provide point-by-point responses to further improve this work as follows.
>
> -----
>
>
> #### **[W1] A theoretical result or comments on the convergence of Algorithms 1 and 2 would make the presentation stronger.**
>
> **Response.** Thank you so much for your insightful suggestion. We agree that adding theoretical comments on the convergence of Algorithms 1 and 2 would strengthen the presentation.
> - Firstly, **we provide theoretical comments on the convergence of Algorithms 1 and 2 based on certain assumptions.** The discussion is structured as follows.
>  - S1. As a preface, we assume that the NNs $g_\theta$ under consideration are convex functions. This assumption aims to bypass the difficulty above. Researchers can satisfy this assumption by employing the convex neural networks in practice.
>  - S2. By Eq. (2), the loss function is a quadratic function, which is convex with respect to the output of the NN $g_\theta(\mathbf{X})$ and correlation matrix $\bar{\boldsymbol{\Sigma}}$.
>  - S3. The composition of convex functions is convex. Integrating this property with S1 and S2, the quadratic loss function is convex with respect to the model parameters $\theta$ and $\bar{\boldsymbol{\Sigma}}$.
>  - S4. Algorithm 1 updates the model parameters $\theta$ with gradient descent, with the correlation matrix $\bar{\boldsymbol{\Sigma}}$ fixed. According to S3, the loss function is convex with respect to the model parameters $\theta$. Therefore, the model parameters $\theta$ will converge, guaranteed by the standard error bound of gradient descent as follows. As the iteration $k$ increases, the error bound decreases, ensuring the convergence of the model parameters $\theta$.
>  - S5. Algorithm 2 updates the correlation matrix $\bar{\boldsymbol{\Sigma}}$ with gradient descent, with the model parameters $\theta$ fixed. Similar to S4, the loss function is convex with respect to the correlation matrix $\bar{\boldsymbol{\Sigma}}$. Therefore, the correlation matrix $\bar{\boldsymbol{\Sigma}}$ will converge, similarly supported by the standard error bound of gradient descent.
>
>  - **Theorem.** Suppose $\theta\_k$ denotes the parameter at the $k$-th iteration, and $\theta^{\star}$ represents the optimal parameter values that minimize $\mathcal{L}(\theta)$. We make three assumptions as follows. (i) The loss function $\mathcal{L}(\theta)$ is convex with respect to $\theta$; (ii) The loss function $\mathcal{L}(\theta)$ has Lipschitz continuous gradients with a Lipschitz constant $L > 0$; (iii) The initial parameter satisfies $\|\theta_0-\theta^{\star}\| \leq \epsilon$ for some constant $\epsilon > 0$. Under these conditions, for all iterations $\mathrm{k} \geq 0$ and learning rate $\eta$, we have $\mathcal{L}(\theta_k)-\mathcal{L}(\theta^{\star}) \leq \frac{1}{2\eta\mathrm{k}}\epsilon^2$ [1].
>
> - Moreover, **we add empirical experiments on the convergence of Algorithms 1 and 2.** Specifically, we visualize the loss curves of Algorithms 1 and 2 in the revised manuscript (Figure 10). Two observations are noted which verify the convergence of Algorithms 1 and 2. (i) The loss curves of both Algorithms 1 and 2 reduce in an approximately monotonic manner, indicating the utility of gradient descent in optimizing the loss function. (ii) The loss curves of both Algorithms 1 and 2 converge to a stable value, indicating the convergence of the loss functions.
> - **Revision.** **We have added the loss curves of Algorithms 1 and 2 in the revised manuscript (Figure 10).**
>
> [1] Boyd, Stephen, and Lieven Vandenberghe. Convex optimization. Cambridge university press, 2004.

---

> ### Author Response · Authors · 2025-11-29
>
> #### **[W2] Emphasizing the above point, the proposed method may be too computationally expensive for some models, given the bilevel optimization, over splits required to perform the optimization to find $\Sigma$**
>
> **Response.** Thank you for your thoughtful comment. **We clarify that the optimization process is highly efficient for the following reasons.**
> - Firstly, the efficiency of the inner-loop optimization is ensured. The procedure involves performing $N_{in}$ steps of gradient descent to update $\theta$, followed by a single gradient descent step to update $\Sigma$. As evidenced by Figure 4, a relatively small value of $N_{in}$ is sufficient to achieve significant performance gains; furthermore, each update step constitutes a standard stochastic gradient descent operation, which is computationally efficient. The update of $\Sigma$ similarly consists of a single, straightforward gradient descent step. Consequently, both the number of iterations and the computational cost per iteration in the inner loop are limited, ensuring the overall efficiency of the inner-loop optimization.
> - Secondly, the efficiency of the outer-loop optimization is ensured. It executes the inner-loop optimization by $K$ times to update $\Sigma$. According to Figure 4, a small value of $K<5$ suffices to achieve leading performance. Consequently, both the number of iterations and the computational cost per iteration are limited, ensuring the overall efficiency of the outer-loop optimization.
> - We add experiments to showcase the actual running time of the inner loop (Algorithm 1) and outer loop (Algorithm 2) in our study. We set $K=3$, which is used in our study and exhibits good performance. The experiments are conducted 10 times with means and standard errors reported. The results demonstrate that even with the largest forecast horizon $T=720$, the computation time of the inner and outer loop is within 2 seconds.
>
>
> | Direction | Loop   | T=64          | T=96          | T=128         | T=192         | T=256         | T=336         | T=512         | T=720         |
> |-----------|--------|---------------|---------------|---------------|---------------|---------------|---------------|---------------|---------------|
> | Forward   | Inner  | 1.196±0.007   | 1.175±0.011   | 1.176±0.009   | 1.409±0.012   | 1.431±0.015   | 1.590±0.011   | 1.523±0.012   | 1.763±0.011   |
> |           | Outer  | 1.161±0.010   | 1.168±0.014   | 1.162±0.009   | 1.350±0.019   | 1.366±0.011   | 1.357±0.013   | 1.361±0.012   | 1.672±0.012   |
> | Backward  | Inner  | 1.147±0.005   | 1.144±0.005   | 1.137±0.006   | 1.487±0.007   | 1.492±0.008   | 1.647±0.007   | 1.591±0.010   | 1.770±0.010   |
> |           | Outer  | 0.954±0.006   | 0.967±0.006   | 0.950±0.008   | 1.190±0.009   | 1.194±0.009   | 1.343±0.008   | 1.296±0.007   | 1.477±0.008   |
>
>
>
> #### **[W3] Some parts of the text are not very clear: For instance, on the results in Section 4.2, which model is used to compare the different forecasting objectives? Similarly in Table 3 (Ablation study).**
>
>  **Response.** We agree that the clarity of the text could be improved. We would like to address this concern as follows.
>
>  - Firstly, we clarify that **in Table 1 and Table 3**, QDF employs the top-performing baseline `TQNet` as the forecasting model. This approach allows us to evaluate whether QDF is able to consistently enhance the performance of the best existing forecasting models.
>
>  - **Revision.** We add a tablenote in Table 1 and 3: `QDF employs the top-performing TQNet as the forecast model.`
>
>
>  #### **[W4] The clarity of the last paragraph of Section 3.2 could be improved.**
>  **Response.** Thank you for your thoughtful suggestion. The paragraph in question outlines the workflow of Algorithm 1. **We have rewritten the paragraph to enhance its clarity and logical flow.** Moreover, **the corresponding step numbers at Algorithm 1 are specially highlighted** within the sentence to help the reader understand the correspondence.

---

> ### Author Response · Authors · 2025-11-29
>
> #### **[W5] The connection of this method with meta-learning is not very clear as it is not formally stated.**
>
> **Response.** Thank you for pointing out the need to clarify the connection between our method (QDF) and meta-learning. Below, we formally summarize both **similarities and differences**.
> - **First, we discuss the similarities with meta-learning:**  Our method (QDF) adopts a bi-level optimization framework that is structurally reminiscent of meta-learning: there is an *inner loop* updating model parameters and an *outer loop* updating a learnable objective component (in our case, the weighting matrix $\Sigma$). This echoes the meta-learning paradigm, where meta-parameters (such as initializations or hyperparameters) are learned by maximizing the induced model's performance.
>
> - **Second, we discuss the key differences with meta-learning** from four aspects. The comparative table is provided below with detailed analysis as follows.
>   - **Different purpose:** Meta-learning is designed to learn a strategy to produce models that can quickly adapt to new or unseen tasks—such as in few-shot or transfer learning. In contrast, QDF focuses on a single, fixed forecasting problem, aiming to address unique issues such as label autocorrelation and heterogeneous task weighting within that problem. Therefore, the purposes are fundamentally different.
>   - **Different object learned:** Meta-learning mainly tunes the model itself or its learning process; QDF instead learns a *parameterization of the loss function*, typically the weighting matrix. Therefore, the typical objects learned are fundamentally different.
>   - **Different validation strategy:** In meta-learning, validation is always over different held-out tasks to measure generalization across tasks. In QDF, the outer loop partitions within the same dataset and task, since we are seeking for an effective weighting matrix for the same forecasting problem. Therefore, the validation strategies are fundamentally different.
>   - **Different output:** Meta-learning produces adaptable initializers or optimizers; QDF produces a loss (typically an optimized weighting matrix $\bar{\Sigma}$) that can be used to better train *any* forecasting model in that task.  Therefore, the algorithm's outputs are fundamentally different.
>
> | Aspect                | Classic Meta-learning                                  | QDF (Ours)                                                            |
> |-----------------------|-------------------------------------------------------|-----------------------------------------------------------------------|
> | **Purpose**      | Fast adaptation to new/unseen tasks  | Addressing label autocorrelation and heterogeneous task weighting challenges.                  |
> | **Learned object**  | Initialized model parameters, optimizers parameters | Weighting matrix $\Sigma$ (to adjust the learning objective)       |
> | **Validation**  | Validating on *different* tasks to ensure task transferability | Validating on a *single* forecasting task to improve in-task performance |
> | **Output**           | Adaptable initializers, optimizers     | A learned objective (typically a learned weighting matrix $\bar{\Sigma}$)                       |
>
> #### **[W6] There's the unmentioned assumption that the time series is non-heteroscedastic and thus is characterized by a single $\Sigma$.**
> **Response.** We agree that this assumption should be mentioned in the paper. Please see our actions as follows.
> - Firstly, we slightly clarify that the assumption is mentioned in the second paragraph of our conclusion section: `A limitation of the current QDF is its reliance on a fixed quadratic objective, parameterized by a static weighting matrix $\bar{\Sigma}$. While being well motivated to address the two challenges, this structure offers limited flexibility.`. It highlights the reliance of QDF on a single $\Sigma$.
> - Moreover, we propose a promising enhancement to address this assumption as a future work: `A promising enhancement would be to employ a hyper-network to generate the weighting matrix $\Sigma$, which yields a more adaptable and expressive formulation, potentially leading to further performance gains.`
> - **Revision.** **We have revised the paragraph for limitation discussion to be more detailed and centered on the assumption.**

---

> ### Author Response · Authors · 2025-11-29
>
> #### **[Q1] Are some architectures more prone to converge to correlated residuals than others?**
> **Response.** Thank you for this insightful question. Based on our analysis, we do not observe a direct or systematic relationship between the choice of model architecture and the  correlated residuals. Our detailed considerations are as follows:
>
> - **The label autocorrelation (correlated residuals) is a property of the label sequences $\mathbf{Y}$**, reflecting temporal dependencies within the labels. Modeling it necessitates modifications to the learning objective, which motivates the design of our QDF objective, a central contribution of this work.
> - **In contrast, model architectures capture dependencies within the input sequences $\mathbf{X}$ and are not equipped to model label autocorrelation in the label sequences**. A dedicated model architecture cannot model the label autocorrelation in $\mathbf{Y}$, since $\mathbf{Y}$ is not fed into the model architectures.
> - **Consequently, modeling label autocorrelation (correlated residuals) hinges on training objective design, unrelated to model architectures.**
>
>
>
> #### **[Q2] In the Ablation study (4.4) how are the two QDFs integrated, by taking their average?**
> **Response.** Yes, the `Avg` column is computed as the average results over forecast horizons: T=96, 192, 336 and 720. We add a footnote in Table 3 to clarify this: `Avg indicates average results over forecast horizons: T=96, 192, 336 and 720.`
>
> Perhaps I missed something, and there is another way to interpret this query, i.e., how the two QDF variants (QDF$^\dagger$ and QDF$^\ddagger$) are implemented. We provide further clarifications as follows to provide a more comprehensive answer.
> - **QDF$^\dagger$** modifies QDF by setting the off-diagonal elements of the weighting matrix $\bar{\Sigma}$ to zero. In this way, it disables the modeling of label autocorrelation effect.
> - **QDF$^\ddagger$** modifies QDF by setting the diagonal elements of the weighting matrix $\bar{\Sigma}$ to one. In this way, it disables the modeling of heterogeneous task weights.
>
> #### **[Q3] Visualizing the correlation matrices of the residuals after training with QDF would be interesting to see.**
> **Response.** Thank you for your actionable suggestion. We agree that visualizing the correlation matrices of residuals is interesting. **We add experiments to visualize them under six settings.** The results are available in Figure 11 in the revised manuscript.

---

### Official Review · Reviewer_LpcK · 2025-11-01

**Soundness:** 3
**Presentation:** 3
**Contribution:** 3
**Rating:** 6
**Confidence:** 2

**Summary:**

The authors propose a method named Quadratic Direct Forecast (QDF) for time series forecasting. The key idea is to optimize the training objective via a quadratic-form weighting matrix to capture both label autocorrelation and weighing different forecasting tasks. The authors apply their proposed method to baseline models and show that it improves the performance in prediction accuracy.

**Strengths:**

1. Overall this paper is clearly written and easy to understand. The technical details are sound and mostly sufficient.

2. The proposed method to improve the training objectives for time series forecasting is novel and applicable to related problems in this domain.

3. The authors provide the source code of their implementation. After reviewing the source code, I did not find major issues.

4. The authors perform rigorous evaluations and helpful ablation studies to understand the impact of different components in the proposed QDF framework.

**Weaknesses:**

1. In addition to MAE and MSE, the authors should evaluate their proposed method with MAPE (mean absolute percentage error) which is robust under different scales of the time series values.

2. The authors should also evaluate their proposed method on standard benchmark datasets for time series forecasting, such as the M4 competition dataset.

**Questions:**

1. How does the model performance change with the dimensionality of the time series?

2. Related to above, is there a curse of dimensionality in the training objective of QDF? If so, how can this issue be effectively addressed?

---

> ### Author Response · Authors · 2025-11-29
>
> Thank you so much for your encouraging support and appreciation of our **writing, novelty, reproducibility, and experiments**. Below are our responses to the specific query raised.
>
> -----
>
>
> #### **[W1] In addition to MAE and MSE, the authors should evaluate their proposed method with MAPE (mean absolute percentage error) which is robust under different scales of the time series values.**
>
> **Response.** Thank you very much for your actionable suggestion. We agree that adding evaluation with MAPE can effectively improve the comprehensiveness of the experimental metrics. We would like to address this concern as follows.
> - **Additional experiment.** We add experiments to evaluate the MAPE of different learning objectives on four datasets. The underlying forecasting model is selected as TQNet and PDF due to their demonstrated competitive performance. The results in the table below indicate that QDF attains superior performance in 7 out of 8 evaluated cases, showing its effectiveness on the MAPE metric.
> - **Revision.** **We have added the MAPE results in Appendix D.8 (Table 10).**
>
> | Dataset | QDF | Time-o1 | FreDF | Koopman | Dilate | Soft-DTW | DF |
> |---|--|-|--|-|--|--|--|
> | **Forecast model: TQNet** |
> | ETTm1 | **2.305** | 2.315 | 2.313 | 2.783 | 2.338 | 2.319 | 2.338 |
> | ETTh1 | **9.619** | 9.697 | 9.875 | 10.488 | 10.036 | 10.283 | 10.290 |
> | ECL | **2.509** | 2.540 | 2.534 | 2.601 | 2.554 | 5.316 | 2.578 |
> | Weather | **3.054** | 3.098 | 3.086 | 3.573 | 3.121 | 3.276 | 3.121 |
> | **Forecast model: PDF** |
> | ETTm1 | **2.232** | 2.420 | 2.412 | 2.876 | 2.421 | 2.391 | 2.283 |
> | ETTh1 | **9.565** | 10.563 | 10.468 | 11.255 | 10.648 | 10.980 | 9.846 |
> | ECL | 2.750 | 2.795 | 2.787 | 2.757 | **2.749** | 5.199 | 2.869 |
> | Weather | **3.156** | 3.226 | 3.189 | 3.293 | 3.244 | 5.064 | 3.228 |
>
>
> #### **[W2] The authors should also evaluate their proposed method on standard benchmark datasets for time series forecasting, such as the M4 competition dataset.**
>
> **Response.** Once again, we express our sincere gratitude for your actionable suggestion. We agree that evaluating the proposed method on additional standard benchmark datasets is beneficial. We would like to address this concern as follows.
> - **Additional experiment.** **We add experiments on the M4 dataset**. The forecasting architectures are selected as TQNet and PDF for their recency and competitive performance. The results in the tables below demonstrate that **QDF performs best in 21 /30 cases, yielding the best overall performance**. Moreover, **QDF consistently outperforms the DF baseline across all cases**, which backs up the improvement of QDF over the prevalent DF approach--the central claim of this paper.
>
> - **Revision.** **We have added the M4 results in Appendix D.9 (Table 11).**
>
>
> | Dataset | QDF ||| Time-o1 ||| FreDF ||| Koopman ||| DF |||
> |---|:-:|:--:|:-:|:---:|:--:|:---:|:---:|:--:|:-:|:---:|:--:|:---:|:--:|:-:|:--:|
> ||SMAPE|MASE|OWA|SMAPE|MASE|OWA|SMAPE|MASE|OWA|SMAPE|MASE|OWA|SMAPE|MASE|OWA|
> |**TQNet**|
> | **Yearly** | 13.355 | 3.015 | 0.788 | **13.377** | **3.004** | **0.787** | 13.404 | 3.022 | 0.790 | 22.588 | 5.512 | 1.385 | 13.502 | 3.074 | 0.800 |
> | **Quarterly** | **10.018** | **1.174** | **0.883** | 10.174 | 1.200 | 0.899 | 10.116 | 1.196 | 0.895 | 17.713 | 2.415 | 1.685 | 10.132 | 1.192 | 0.895 |
> | **Monthly** | **12.756** | **0.939** | **0.884** | 12.776 | 0.949 | 0.889 | 12.786 | 0.952 | 0.891 | 18.655 | 1.506 | 1.355 | 12.777 | 0.945 | 0.887 |
> | **Others** | 4.909 | 3.203 | **1.022** | 5.039 | 3.285 | 1.048 | **4.908** | **3.219** | 1.024 | 7.478 | 5.365 | 1.633 | 5.048 | 3.292 | 1.050 |
> | **Average** | **11.844** | **1.586** | **0.851** | 11.903 | 1.599 | 0.857 | 11.894 | 1.600 | 0.857 | 18.775 | 2.839 | 1.434 | 11.923 | 1.611 | 0.861 |
> |**PDF**|
> ||SMAPE|MASE|OWA|SMAPE|MASE|OWA|SMAPE|MASE|OWA|SMAPE|MASE|OWA|SMAPE|MASE|OWA|
> | **Yearly** | 13.426 | 3.044 | 0.794 | **13.524** | **3.014** | **0.793** | 13.479 | 3.052 | 0.796 | 23.515 | 5.695 | 1.436 | 13.532 | 3.036 | 0.796 |
> | **Quarterly** | **10.361** | **1.224** | **0.917** | 10.690 | 1.282 | 0.953 | 10.367 | 1.241 | 0.923 | 19.090 | 2.572 | 1.804 | 10.646 | 1.279 | 0.950 |
> | **Monthly** | **12.930** | **0.961** | **0.900** | 13.181 | 1.003 | 0.928 | 13.023 | 0.987 | 0.916 | 20.595 | 1.756 | 1.540 | 13.208 | 0.999 | 0.928 |
> | **Others** | **4.891** | 3.262 | **1.029** | 5.012 | **3.256** | 1.041 | 5.381 | 3.579 | 1.130 | 9.890 | 8.213 | 2.336 | 5.698 | 3.735 | 1.188 |
> | **Average** | **12.026** | **1.618** | **0.866** | 12.254 | 1.645 | 0.882 | 12.108 | 1.653 | 0.879 | 20.370 | 3.181 | 1.583 | 12.292 | 1.672 | 0.890 |

---

> ### Author Response · Authors · 2025-11-29
>
> #### **[Q1] How does the model performance change with the dimensionality of the time series?**
>
> **Response.** Thank you very much for your insightful comment. We are pleased to discuss the impact of dimensionality on the performance of QDF. specifically, **there are two types of dimensionality in this work**: (i) the forecast horizon (**temporal dimensionality**) and (ii) the number of features to forecast (**covariate dimensionality**). We discuss both as follows.
> - Firstly, we discuss the impact of the **temporal dimensionality.** In Table 6, we include the forecasting performance with different forecast horizons. The results show that **increasing temporal dimensionality negatively impacts the forecasting performance.** This is a natural observation, as more distant future steps are more difficult to predict, and increasing the number of tasks brings optimization challenges (e.g., the contradictory gradient directions among different tasks).
> - Secondly, we discuss the impact of the **covariate dimensionality.** To this end, **we add experiments: we mask to preserve 30%, 50%, 70%, and 100% of the features in the dataset**, train the forecasting model on the masked dataset, and evaluate the forecasting performance. The results are available in the table below, where $^\dagger$ denotes the forecasting models trained with QDF. There are two primary observations: (i) In most cases, **increasing the covariate dimensionality negatively impacts the forecasting performance.** The rationale is similar to the temporal dimensionality: involving more covariates increases the number of tasks and thus complicating the optimization process. (ii) **QDF consistently outperforms the DF baseline** across different feature ratios, which further validates its utility and robustness.
>
>
> | Dataset | Feature Ratio | TQNet$^\dagger$ | | TQNet | | PDF$^\dagger$ | | PDF | |
> |---|--|----|---|---|---|---|--|-|-|
> | | |MSE|MAE|MSE|MAE|MSE|MAE|MSE|MAE|
> | **ETTm1** | 0.3 | 0.091 | 0.222 | 0.093 | 0.224 | 0.093 | 0.223 | 0.095 | 0.225 |
> | | 0.5 | 0.239 | 0.326 | 0.242 | 0.329 | 0.236 | 0.326 | 0.250 | 0.335 |
> | | 0.7 | 0.223 | 0.321 | 0.228 | 0.325 | 0.229 | 0.328 | 0.238 | 0.335 |
> | | 1.0 | 0.371 | 0.389 | 0.376 | 0.391 | 0.381 | 0.394 | 0.387 | 0.396 |
> | **Weather** | 0.3 | 0.235 | 0.171 | 0.238 | 0.174 | 0.244 | 0.183 | 0.253 | 0.183 |
> | | 0.5 | 0.254 | 0.210 | 0.256 | 0.212 | 0.273 | 0.225 | 0.274 | 0.222 |
> | | 0.7 | 0.231 | 0.231 | 0.233 | 0.233 | 0.246 | 0.243 | 0.249 | 0.242 |
> | | 1.0 | 0.242 | 0.268 | 0.246 | 0.270 | 0.259 | 0.281 | 0.265 | 0.283 |
>
>
> #### **[Q2] Related to above, is there a curse of dimensionality in the training objective of QDF? If so, how can this issue be effectively addressed?**
>
> **Response.** Thank you very much for your meticulous comment--it seems to be a follow-up
> question of [Q1]. We are pleased to further discuss the curse of dimensionality.
> - Firstly, **we would suggest focusing on the temporal dimensionality for discussion**. While
> both temporal and covariate dimensionality can contribute to the curse of dimensionality, QDF
> is mainly related to temporal dimensionality due to its emphasis on label autocorrelation. The
> impact of covariate dimensionality, on the other hand, is not unique to QDF and is not the
> main focus of this work. Therefore, we focus on the temporal dimensionality for discussion.
> - Secondly, **we add experiments to test for the curse of dimensionality** in QDF. The only learnable parameter introduced by QDF is the covariance matrix $\boldsymbol{\Sigma} \in \mathbb{R}^{T \times T}$, whose size grows with the temporal dimension $T$. To examine whether the curse of dimensionality affects the performance, we varied the rank of $\boldsymbol{\Sigma}$—and thus the number of its free parameters—by constraining it to different proportions of its full rank. The performance is reported in the table below. Overall, **changing the rank of $\boldsymbol{\Sigma}$ has little effect on QDF's performance, implying that the curse of dimensionality is not a significant concern in our scenario.** Nonetheless, if dimensionality ever becomes an issue, employing a low-rank approximation for $\boldsymbol{\Sigma}$ is an effective way to reduce parameters and alleviate such problems.

---

> ### Author Response · Authors · 2025-11-29
>
> | Dataset | Horizon | 100% rank || 80% rank || 60% rank || 40% rank || 20% rank || DF ||
> |--|-|--|--|--|--|--|--|--|--|--|--|-|-|
> | **TQNet** ||MSE|MAE|MSE|MAE|MSE|MAE|MSE|MAE|MSE|MAE|MSE|MAE|
> | ETTm1 | 96 | 0.307 | 0.349 | 0.308 | 0.350 | 0.310 | 0.351 | 0.309 | 0.351 | 0.310 | 0.352 | 0.310 | 0.352 |
> | | 192 | 0.352 | 0.376 | 0.354 | 0.377 | 0.355 | 0.379 | 0.354 | 0.378 | 0.352 | 0.377 | 0.356 | 0.377 |
> | | 336 | 0.383 | 0.398 | 0.387 | 0.399 | 0.387 | 0.399 | 0.387 | 0.400 | 0.387 | 0.401 | 0.388 | 0.400 |
> | | 720 | 0.441 | 0.434 | 0.445 | 0.437 | 0.446 | 0.437 | 0.443 | 0.437 | 0.445 | 0.437 | 0.450 | 0.437 |
> | | Avg | 0.371 | 0.389 | 0.373 | 0.391 | 0.375 | 0.392 | 0.373 | 0.391 | 0.373 | 0.392 | 0.376 | 0.391 |
> | Weather | 96 | 0.158 | 0.201 | 0.158 | 0.201 | 0.159 | 0.202 | 0.158 | 0.201 | 0.159 | 0.202 | 0.160 | 0.203 |
> | | 192 | 0.207 | 0.245 | 0.208 | 0.246 | 0.208 | 0.246 | 0.207 | 0.246 | 0.207 | 0.246 | 0.210 | 0.247 |
> | | 336 | 0.263 | 0.286 | 0.264 | 0.287 | 0.264 | 0.287 | 0.263 | 0.287 | 0.264 | 0.287 | 0.267 | 0.289 |
> | | 720 | 0.342 | 0.339 | 0.342 | 0.339 | 0.342 | 0.339 | 0.342 | 0.339 | 0.342 | 0.340 | 0.346 | 0.342 |
> | | Avg | 0.242 | 0.268 | 0.243 | 0.268 | 0.243 | 0.269 | 0.243 | 0.268 | 0.243 | 0.269 | 0.246 | 0.270 |
> | **PDF** |||||||||||||||
> | ETTm1 | 96 | 0.320 | 0.358 | 0.314 | 0.355 | 0.315 | 0.353 | 0.317 | 0.358 | 0.314 | 0.356 | 0.326 | 0.363 |
> | | 192 | 0.361 | 0.380 | 0.358 | 0.378 | 0.358 | 0.376 | 0.359 | 0.379 | 0.361 | 0.380 | 0.365 | 0.381 |
> | | 336 | 0.390 | 0.401 | 0.389 | 0.401 | 0.389 | 0.402 | 0.389 | 0.399 | 0.389 | 0.398 | 0.397 | 0.402 |
> | | 720 | 0.451 | 0.437 | 0.449 | 0.434 | 0.447 | 0.434 | 0.448 | 0.435 | 0.447 | 0.435 | 0.458 | 0.437 |
> | | Avg | 0.381 | 0.394 | 0.377 | 0.392 | 0.377 | 0.391 | 0.378 | 0.393 | 0.378 | 0.392 | 0.387 | 0.396 |
> | Weather | 96 | 0.176 | 0.218 | 0.178 | 0.219 | 0.181 | 0.221 | 0.179 | 0.220 | 0.179 | 0.220 | 0.181 | 0.221 |
> | | 192 | 0.225 | 0.260 | 0.226 | 0.260 | 0.227 | 0.261 | 0.226 | 0.260 | 0.227 | 0.260 | 0.232 | 0.262 |
> | | 336 | 0.280 | 0.299 | 0.280 | 0.298 | 0.281 | 0.298 | 0.279 | 0.299 | 0.280 | 0.299 | 0.285 | 0.300 |
> | | 720 | 0.357 | 0.347 | 0.357 | 0.347 | 0.357 | 0.347 | 0.357 | 0.348 | 0.356 | 0.347 | 0.360 | 0.348 |
> | | Avg | 0.259 | 0.281 | 0.260 | 0.281 | 0.261 | 0.282 | 0.260 | 0.282 | 0.260 | 0.282 | 0.265 | 0.283 |

---

### Author Response · Authors · 2025-12-02

Dear AC and all reviewers,

We sincerely appreciate your great efforts in evaluating our paper despite your busy schedules. We are encouraged that all reviewers are in the positive side, with scores 6, 6, and 6, recognizing our paper
- ``is novel and applicable to related problems in this domain’’ (Reviewer LpcK),
- ``introduces a technically novel strategy to solve an empirically and theoretically motivated problem’’ (Reviewer VKq3),
- with experiments ``effectively presented and structured to support the paper's claims’’ (Reviewer wfuZ).

We also value the constructive feedback provided, helping us improve the quality of our manuscript. In response, we have incorporated the following editions into our newly-uploaded manuscript:
- **Response to Reviewer LpcK:** We have added experimental evaluation on the MAPE metric (Table 10) and the M4 dataset (Table 11). Also, we conducted experiments demonstrating that QDF does not substantially suffers from the curse of dimensionality.
- **Response to Reviewer VKq3:** We added both theoretical and empirical evidence of algorithm convergence and showcased efficiency, and we have clarified the relationship between QDF and meta-learning. Furthermore, we explicitly stated and discussed the non-heteroscedastic assumption and added visualizations of the correlation matrices.
- **Response to Reviewer wfuZ:** We theoretically formalized the advantage of QDF over FreDF and Time-o1, demonstrating that QDF acts as a generalization of these methods. We also clarified our data splitting strategy to confirm the absence of data leakage. Finally, we have refined the code repository to enhance reproducibility and discussed the potential and limitations of applying QDF to other multitask scenarios.

We believe these revisions address the current concerns raised and effectively strengthen our manuscript. We would highly appreciate it if you could kindly consider this context when making your recommendation.

Thank you in advance,

Authors of 25573

---

### Meta-Review · Area_Chair_GP57 · 2026-01-07

**Summary:**

The authors propose a method named Quadratic Direct Forecast (QDF) for time series forecasting. The key idea is to optimize the training objective via a quadratic-form weighting matrix to capture both label autocorrelation and weighing different forecasting tasks. Three reviewers provided positive feedbacks on this paper. Concerns were mostly resolved in the rebuttal phase, including extra evaluation metrics and datasets, theoretical analysis, etc. Therefore, I recommend acceptance of this paper.

**Reviewer Concerns:**

Reviewer LpcK:

W1: MAPE results. Fully addressed.
W2: M4 results. Fully addressed.
Q1&2: Model behaviors w.r.t.  dimensionality of the time series. Partially addressed. The authors provided some empirical results.

Reviewer VKq3:

W1: Theoretical analysis. Partially addressed. The authors provided some intuitions but no formal proof.
W2: Efficiency. Partially addressed. The authors report running time of inner and outer training loops, but not provide efficiency comparison with baselines.
W3-5: Clarification questions. Fully addressed.

Reviewer wfuZ:

W1: Theoretical analysis on weighting matrix. Partially addressed. The authors provided some intuitions but no formal proof.
W2: Information leakage. Fully addressed.
W3: Code and reproducibility. Fully addressed. Authors should ensure code is publicly available with clear README instructions upon acceptance.

**Reviewer Scores:**

Given all reviewers' positive feedbacks and partially resolved concerns, I think the reviewers' scores will remain the same.

---

### Decision · Program_Chairs · 2026-01-26

Accept (Poster)